# SMARCAD1 and TOPBP1 contribute to heterochromatin maintenance at the transition from the 2C-like to the pluripotent state

Ruben Sebastian-Perez[1†], Shoma Nakagawa[1†], Xiaochuan Tu[1†], Sergi Aranda[1], Martina Pesaresi[1], Pablo Aurelio Gomez-Garcia[1], Marc Alcoverro-Bertran[1], Jose Luis Gomez-Vazquez[1], Davide Carnevali[1], Eva Borràs[1,2], Eduard Sabidó[1,2], Laura Martin[1], Malka Nissim-Rafinia[3], Eran Meshorer[3,4], Maria Victoria Neguembor[1], Luciano Di Croce[1,2,5], Maria Pia Cosma[1,2,5,6]*

[1]Centre for Genomic Regulation (CRG), The Barcelona Institute of Science and Technology, Barcelona, Spain; [2]Universitat Pompeu Fabra, Barcelona, Spain; [3]Department of Genetics, The Alexander Silberman Institute of Life Sciences, Edmond J. Safra Campus, The Hebrew University of Jerusalem, Jerusalem, Israel; [4]The Edmond and Lily Safra Center for Brain Sciences, Edmond J. Safra Campus, The Hebrew University of Jerusalem, Jerusalem, Israel; [5]ICREA, Barcelona, Spain; [6]Medical Research Institute, Guangdong Provincial People's Hospital (Guangdong Academy of Medical Sciences), Southern Medical University, Guangzhou, China

*For correspondence:
pia.cosma@crg.es

[†]These authors contributed equally to this work

Competing interest: The authors declare that no competing interests exist.

## eLife Assessment

This **important** study examines heterochromatin domain dynamics using a model system that allows reversible transition from an embryonic stem cell to a 2-cell-like state. The authors present a **solid** resource to the research community that will further the understanding of changes in the chromatin-bound proteome during the 2C-to-ESC transition. However, conclusions related to the functional roles of the interaction between the SWI/SNF complex component SMARCAD1 and the DNA Topoisomerase II Binding protein (TOPBP1) remain incomplete.

**Abstract** Chromocenters are established after the 2-cell (2C) stage during mouse embryonic development, but the factors that mediate chromocenter formation remain largely unknown. To identify regulators of 2C heterochromatin establishment in mice, we generated an inducible system to convert embryonic stem cells (ESCs) to 2C-like cells. This conversion is marked by a global reorganization and dispersion of H3K9me3-heterochromatin foci, which are then reversibly formed upon re-entry into pluripotency. By profiling the chromatin-bound proteome (chromatome) through genome capture of ESCs transitioning to 2C-like cells, we uncover chromatin regulators involved in de novo heterochromatin formation. We identified TOPBP1 and investigated its binding partner SMARCAD1. SMARCAD1 and TOPBP1 associate with H3K9me3-heterochromatin in ESCs. Interestingly, the nuclear localization of SMARCAD1 is lost in 2C-like cells. SMARCAD1 or TOPBP1 depletion in mouse embryos leads to developmental arrest, reduction of H3K9me3, and remodeling of heterochromatin foci. Collectively, our findings contribute to comprehending the maintenance of chromocenters during early development.

## Introduction

Early mammalian development is a dynamic process that involves large-scale chromatin reorganization (*Burton and Torres-Padilla, 2014*). Blastomeres acquire a defined cell identity through the activation of a subset of genes and specific epigenetic modifications. Among the latter, there is a tailored control over histone H3 lysine 9 trimethylation (H3K9me3), a hallmark of the transcriptionally repressed constitutive heterochromatin (*Nakayama et al., 2001*; *Peters et al., 2001*). During the first cleavage stages, constitutive heterochromatin reorganizes in the nucleus to form highly compacted chromocenters (*Probst and Almouzni, 2011*; *Jones, 1970*). Systematic identification of the underlying factors involved in de novo heterochromatin establishment and maintenance - thus chromocenter compaction - is still lacking, mostly because of the minuscule amount of material available during embryogenesis.

Embryonic stem cells (ESCs) can fluctuate back to a 2-cell embryo-like (2C-like) state under defined culture conditions (*Macfarlan et al., 2012*). Although ESCs can spontaneously revert their fate to resemble early embryogenesis, this process happens at a very low frequency (*Macfarlan et al., 2012*). Recently, early mouse embryo development has been modeled with high efficiency after downregulation of chromatin assembly factors (*Ishiuchi et al., 2015*) or modulation of key developmentally regulated genes in ESCs (*Hu et al., 2020*; *Eckersley-Maslin et al., 2019*; *De Iaco et al., 2019*; *Hendrickson et al., 2017*). ESCs can efficiently be converted into 2C-like cells by the overexpression of a single murine transcription factor, Dux (*Hendrickson et al., 2017*). Interestingly, decondensation of HP1α foci and of chromocenters in $2C^+$ were previously reported, suggesting that $2C^+$ cells can be used to study the remodeling of heterochromatin foci (*Ishiuchi et al., 2015*; *Akiyama et al., 2015*). Here, using the Dux-dependent reprogramming system, we show that 2C-like cells can be used as a model system to investigate de novo chromocenter formation and dynamics. Using chromatin proteomics, we profiled the dynamic changes occurring in the chromatin-bound proteome (chromatome) during 2C-like cell reprogramming and identified factors potentially involved in chromocenter reorganization. H3K9me3-marked heterochromatin foci in 2C-like cells generated via Dux overexpression became larger and decreased in number during the reprogramming of ESCs to 2C-like cells. The chromocenters re-formed upon transition of 2C-like cells into ESC-like cells. We identified the DNA TOPoisomerase II Binding Protein 1 (TOPBP1) and the chromatin remodeler SWI/ SNF-Related, Matrix-Associated Actin-Dependent Regulator Of Chromatin, Subfamily A, Containing DEAD/H Box 1 (SMARCAD1) to be associated with H3K9me3 in heterochromatin foci of ESCs. The association of SMARCAD1 was reduced upon entry of ESCs in the 2C-like state, although SMARCAD1 nuclear localization was recovered after 2C-like state exit. Depletion of SMARCAD1 and of TOPBP1 induced mouse embryo developmental arrest, which was accompanied by a remodeling of the heterochromatin foci. Our results suggest a contributing role of SMARCAD1 and of TOPBP1 activity in the maintenance of heterochromatin formation during early development.

## Results

### Entry in the 2C-like state is characterized by the remodeling of H3K9me3 heterochromatic regions

To explore the molecular driving events for the establishment of constitutive heterochromatin during mouse embryo development, we generated stable ESC lines carrying doxycycline-inducible cassettes that drive expression of either Dux (Dux-codon altered, CA) or luciferase (control) (*Figure 1A*). These ESC lines also carry an EGFP reporter under the control of the endogenous retroviral element MERVL long terminal repeat (2C::EGFP) (*Ishiuchi et al., 2015*). The EGFP reporter allows the purification of 2C-like cells (hereinafter named $2C^+$) and low *Dux* expressing cells, which are negative for MERVL reporter expression ($2C^-$) (*Figure 1A*). $2C^-$ cells were reported to be an intermediate population generated during 2C-like reprogramming (*Fu et al., 2019*; *Rodriguez-Terrones et al., 2018*). During the reprogramming process, in contrast to $2C^-$ cells, $2C^+$ cells do not show DAPI-dense chromocenters (*Ishiuchi et al., 2015*; *Hendrickson et al., 2017*). Therefore, we can study de novo chromocenter formation by following the transition of $2C^+$ cells toward an ESC-like state, thus modeling in culture the epigenetic reprogramming that occurs during mouse early development.

After culturing the Dux-CA line with doxycycline (Dox), the number of 2C-like cells increased to >60% compared with luciferase control cells (*Figure 1—figure supplement 1A–D*). Dux overexpression

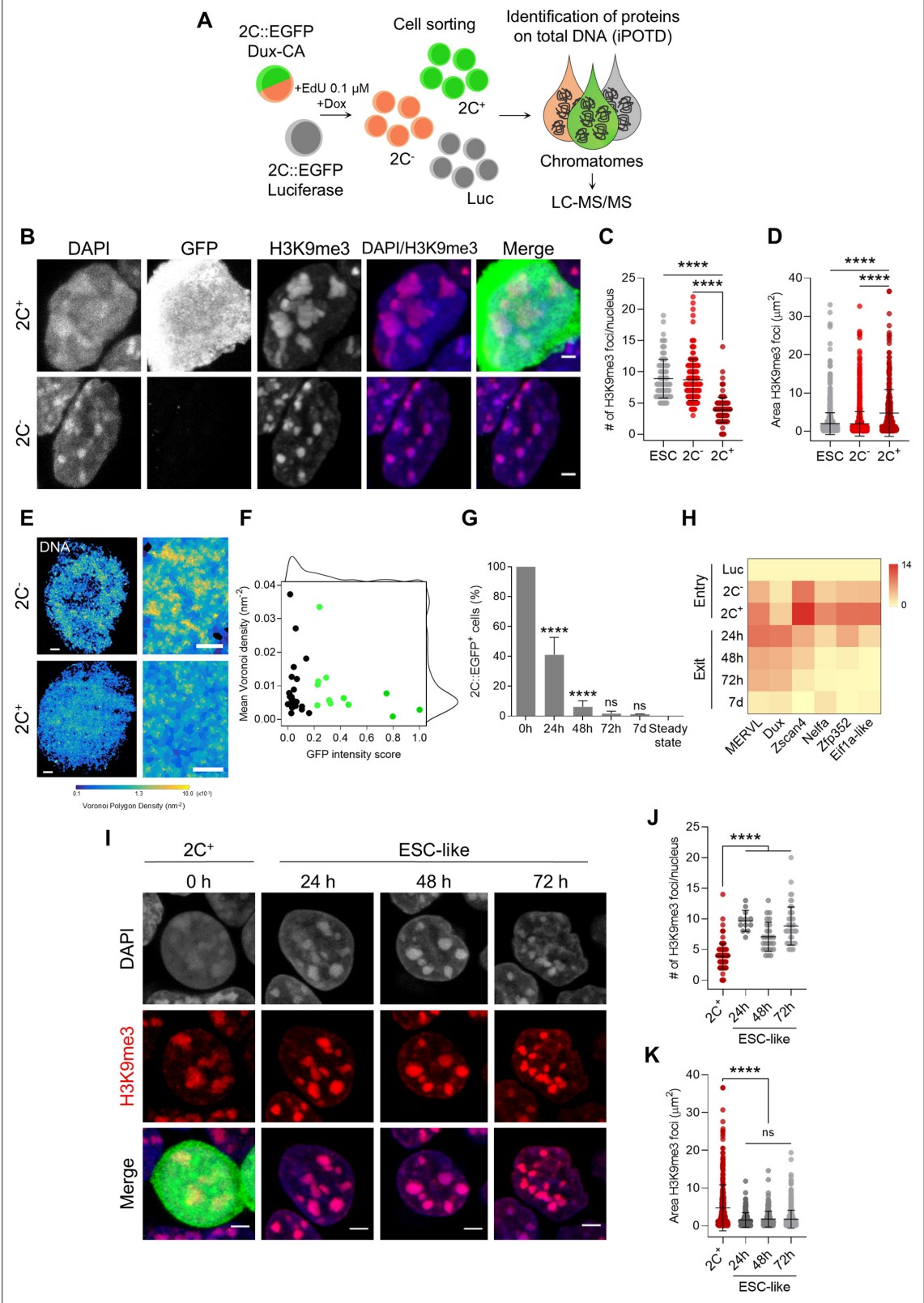

**Figure 1.** Entry in the 2C-like state is characterized by the remodeling of H3 lysine 9 trimethylation (H3K9me3) heterochromatin, which is reverted upon 2C+ exit. (**A**) Schematic representation of the samples collected to perform the identification of protein on total DNA (iPOTD) workflow. LC-MS/MS, liquid chromatography-tandem mass spectrometry. (**B**) Representative immunofluorescence images of the 2C::EGFP reporter and H3K9me3 in 2C- and 2C+ cells. Scale bar, 2 μm. (**C**) Quantification of the number of H3K9me3 foci in embryonic stem cells (ESCs), 2C- and 2C+ cells. Data are

*Figure 1 continued on next page*

*Figure 1 continued*

presented as scatter dot plots with line at mean ± SD (n>3 independent cultures, ESCs=103 cells, 2C⁻=170 cells, and 2C⁺=119 cells). p<0.0001**** by one-way ANOVA (Tukey's multiple comparisons test). (**D**) Quantification of H3K9me3 foci area in ESCs, 2C⁻ and 2C⁺ cells. Data are presented as scatter dot plots with line at mean ± SD (n>3 independent cultures, ESCs=1712 foci, 2C⁻=1445 foci, and 2C⁺=340 foci). p<0.0001**** by one-way ANOVA (Tukey's multiple comparisons test). (**E**) Voronoi tessellation rendering of super-resolution images of DNA in 2C⁻ and 2C⁺ cells. Full nuclei (left; scale bar, 1 μm) and zoomed images (right; scale bar, 400 nm) are shown. (**F**) Biaxial density plot showing mean Voronoi density of DNA (inverse of the polygon area) as a measure of chromatin compaction and GFP intensity score in 2C⁻ and 2C⁺ cells. Cells with a GFP intensity score >0.2 are colored in green. Black dots indicate 2C⁻ cells and green dots indicate 2C⁺ cells. Each dot represents a single cell (2C⁻=23 cells and 2C⁺=12 cells). (**G**) Quantification of the percentage of 2C-like cells 24 hr, 48 hr, 72 hr, and 7 days after 2C⁺ cell sorting. The endogenous 2C-like fluctuation was used as the steady-state condition. Data are presented as mean ± SD (n=3 independent experiments). p=0.7656ⁿˢ, p<0.0001**** by one-way ANOVA (Tukey's multiple comparisons test). (**H**) Heat map representation of *MERVL*, *Dux*, *Zscan4*, *Nelfa*, *Zfp352*, and *Eif1a-like* expression in luciferase (Luc), 2C⁻ and 2C⁺ sorted cells (entry) and in ESC-like cells at 24 hr, 48 hr, 72 hr, and 7 days (7d) after 2C⁺ sorting. Data are presented as log₂ fold change (FC) values to luciferase detected by quantitative real-time PCR (qRT-PCR). (**I**) Representative immunofluorescence images of H3K9me3 at 0 hr (2C⁺ before exit), 24 hr, 48 hr, and 72 hr after 2C-like state exit. Scale bar, 3 μm. (**J**) Quantification of the number of H3K9me3 foci in 2C⁺ cells and at 24 hr, 48 hr, and 72 hr after 2C-like state exit. Data are presented as scatter dot plots with line at mean ± SD (n=2 independent cultures, 2C⁺=119 cells, same dataset plotted in **B**; ESC-like 24 hr=12 cells; ESC-like 48 hr=27 cells; ESC-like 72 hr=49 cells). p<0.0001**** by one-way ANOVA (Tukey's multiple comparisons test). (**K**) Quantification of H3K9me3 foci area in 2C⁺ cells and at 24 hr, 48 hr, and 72 hr after 2C-like state exit. Data are presented as scatter dot plots with line at mean ± SD (n=2 independent cultures, 2C⁺=340 foci, same dataset plotted in **C**; ESC-like 24 hr=168 foci; ESC-like 48 hr=238 foci; ESC-like 72 hr=605 foci). p>0.05ⁿˢ, p<0.0001**** by one-way ANOVA (Tukey's multiple comparisons test).

The online version of this article includes the following figure supplement(s) for figure 1:

**Figure supplement 1.** Characterization of Dux-derived 2C-like cells.

resulted in the decompaction of DAPI-dense chromocenters and loss of the pluripotency transcription factor OCT4 (*Figure 1—figure supplement 1E*), in accordance with previous reports (*Macfarlan et al., 2012*; *Ishiuchi et al., 2015*). These changes were accompanied by an upregulation of specific genes of the 2-cell transcriptional program such as endogenous *Dux*, MERVL, and major satellites (MajSat) (*Figure 1—figure supplement 1F*). Additionally, we looked at cell cycle progression in the heterogeneous population of cells generated after Dux overexpression since it has been previously shown that spontaneous 2C-like cells have an altered cell cycle (*Eckersley-Maslin et al., 2016*). 2C⁻ cells displayed a cell cycle profile comparable to that of luciferase cells, whereas 2C⁺ cells accumulated in the G2/M cell cycle phase (*Figure 1—figure supplement 1G*) with a much-reduced S phase consistent in several clonal lines (*Figure 1—figure supplement 1H*). Overall, these data indicate that the 2C⁺ line we generated recapitulates known features of 2C-like cells.

To study remodeling of the chromocenters, we asked about the reorganization of heterochromatic regions upon reprogramming of ESCs into 2C⁺ cells. H3K9me3 is a well-known pericentric heterochromatin histone modification that prominently associates with constitutive heterochromatin (*Nakayama et al., 2001*; *Peters et al., 2001*; *Peters et al., 2003*; *Rea et al., 2000*). H3K9me3 can therefore be used as a marker for chromocenters. H3K9me3 foci in 2C⁺ cells were morphologically distinct from those of 2C⁻ cells (*Figure 1B*). They were 2.3-fold fewer (3.89±0.19 foci/nucleus) (*Figure 1C*), and occupied 2.4-fold larger area (4.76±0.33 μm²) in 2C⁺ compared with both ESCs (8.88±0.30 foci/nucleus; 1.99±0.07 μm²) and 2C⁻ cells (8.72±0.25 foci/nucleus; 1.93±0.08 μm²) (*Figure 1D*). These results suggest that H3K9me3 heterochromatin undergoes massive spatial reorganization during the reprogramming of ESCs into 2C-like state. Importantly, the levels of H3K9me3 remain unchanged among ESCs, 2C⁻ and 2C⁺ cells, indicating that the remodeling of chromocenters was not due to loss of H3K9me3 (*Figure 3—figure supplement 1D*). The increased size of the H3K9me3 foci and the reduction in the number of H3K9me3 foci per nucleus might be due to the decompaction or fusion of several chromocenters.

We then imaged global DNA organization with Stochastic Optical Reconstruction super-resolution Microscopy (STORM). DNA was labeled using the nucleotide analogue 5-ethynyl-2'-deoxycytidine (EdC) (*Otterstrom et al., 2019*; *Zessin et al., 2012*). DNA images were quantified by Voronoi tessellation analysis (*Andronov et al., 2016*; *Levet et al., 2015*), which can precisely determine the DNA density based on the number of localizations in each Voronoi tessel (see Materials and methods). Voronoi analysis showed a marked decrease in the localization density of the chromatin in 2C⁺ cells (*Figure 1E*). Furthermore, Voronoi analysis confirmed the decreased DNA density as a function of the GFP intensity in 2C⁺ cells (*Figure 1F*). Interestingly, 2C⁻ cells were heterogeneous with respect to DNA density, with the majority of them showing low DNA density compared with 2C⁺ cells, suggesting that

DNA might undergo decompaction prior to GFP activation (*Figure 1F*). Overall, the DNA decompaction of the chromatin fibers in 2C$^+$ cells is consistent with the chromatin landscape of early/late 2-cell embryos, which has been reported to be in a relaxed chromatin state and more accessible, as shown by Assay for Transposase Accessible Chromatin with high-throughput sequencing (ATAC-seq) (*Hendrickson et al., 2017*; *Wu et al., 2016*; *Zhu et al., 2021*).

## H3K9me3 heterochromatin becomes rapidly formed following exit from the 2C-like state

We then asked whether 2C$^+$ cells could undergo the reverse transition, exiting the 2C-like state and subsequently re-entering pluripotency, thereby becoming ESC-like cells. We defined ESC-like cells as those that, after being purified as 2C$^+$ cells, no longer express the MERVL reporter during the exit phase. To answer this question about the kinetics of the reverse transition, we followed the expression of EGFP in fluorescence-activated cell sorting (FACS)-sorted 2C$^+$ cells 24 hr, 48 hr, 72 hr, and 1 week after sorting (*Figure 1G*). Strikingly, over 60% of the 2C$^+$ cells in culture lost the expression of the MERVL reporter 24 hr after sorting. Moreover, 48 hr after sorting, only 6% of the cells still expressed the reporter, suggesting rapid repression of the 2C program and quick re-establishment of the pluripotency network (*Figure 1G*). 72 hr and 7 days after sorting, EGFP expression levels were comparable to those derived from the endogenous fluctuation ('*steady state*') of ESC cultures (*Figure 1G*). The decay in EGFP levels was accompanied by a downregulation of MERVL, endogenous *Dux*, *Zscan4*, *Nelfa*, *Zfp352*, and *Eif1a-like* gene expression (*Figure 1H*). These results indicate that 2C-like cells could revert their fate back to pluripotency after Dux overexpression, and that such transition occurs rapidly, as early as 24 hr after sorting.

We then quantified the number and area of H3K9me3 foci during the 2C$^+$ to ESC-like transition (*Figure 1I–K*). Our results indicate that chromocenters underwent rapid re-formation and increased in number (24 hr: 9.67±0.50; 48 hr: 7.07±0.46; 72 hr: 8.82±0.44 foci/nucleus) compared with 2C$^+$ cells (3.89±0.19 foci/nucleus), concomitantly to the loss of EGFP expression and to the exit from the 2C-like state (*Figure 1I and J*). The areas of chromocenters in ESC-like cells were similar across the different time points analyzed (24 hr: 1.54±0.15; 48 hr: 1.77±0.14; 72 hr: 1.75±0.10 μm$^2$) and smaller of those of 2C$^+$ cells (4.76±0.33 μm$^2$) (*Figure 1I and K*). These results suggest that the in vitro transition of 2C$^+$ cells toward the ESC-like state can be used as a model system to study chromocenter formation and chromatin reorganization occurring during early development.

## Chromatin-bound proteome profiling allows the identification of dynamic chromatome changes during 2C-like cell reprogramming

Having characterized the Dux-CA line, we aimed to identify potential chromatin-associated factors involved in the de novo establishment of heterochromatin. For that, we performed DNA-mediated chromatin purification coupled to tandem mass spectrometry for the identification of proteins on total DNA (iPOTD) (*Aranda et al., 2020*; *Aranda et al., 2019*). We captured the whole genome labeled with 5-ethynyl-2′-deoxyuridine (EdU) and identified candidate proteins differentially enriched in the 2C-like chromatin-bound (chromatome) fraction (*Figure 1A*). We analyzed the chromatome of 2C$^+$, 2C$^-$, and luciferase (Luc) populations to characterize the chromatin-bound proteome profile of these distinct states. We first confirmed that we could enrich the iPOTD preparations for chromatin proteins, such as histone H3, and devoid them of cytoplasmic ones, such as vinculin (*Figure 2—figure supplement 1A–C*). We identified a total of 2396 proteins, suggesting an effective pull-down of putative chromatin-associated factors (*Figure 2—figure supplement 1D* and *Supplementary file 1*). Chromatin-resident proteins, such as core histones and histone variants, were comparably enriched in all +EdU replicates (*Figure 2—figure supplement 1E* and *Supplementary file 1*). Pearson's correlation coefficients (PCC) and principal component analysis of independent replicates of 2C$^+$, 2C$^-$, and Luc samples showed consistent results regarding the abundance of the proteins detected (*Figure 2A–C* and *Supplementary file 1*). Interestingly, Luc replicates clustered separately from 2C$^+$ and 2C$^-$ conditions, indicating significant changes in the chromatomes of these fractions (*Figure 2B and C*).

We then ranked the identified chromatin-associated factors according to their fold change to interrogate the differences in protein-chromatin interactions in the 2C$^+$, 2C$^-$, and Luc chromatomes (*Figure 2D–F*). Members of the ZSCAN4 (zinc finger and SCAN domain containing 4) family of proteins, which are well-characterized markers of the 2C stage (*Ishiuchi et al., 2015*; *Hendrickson et al., 2017*;

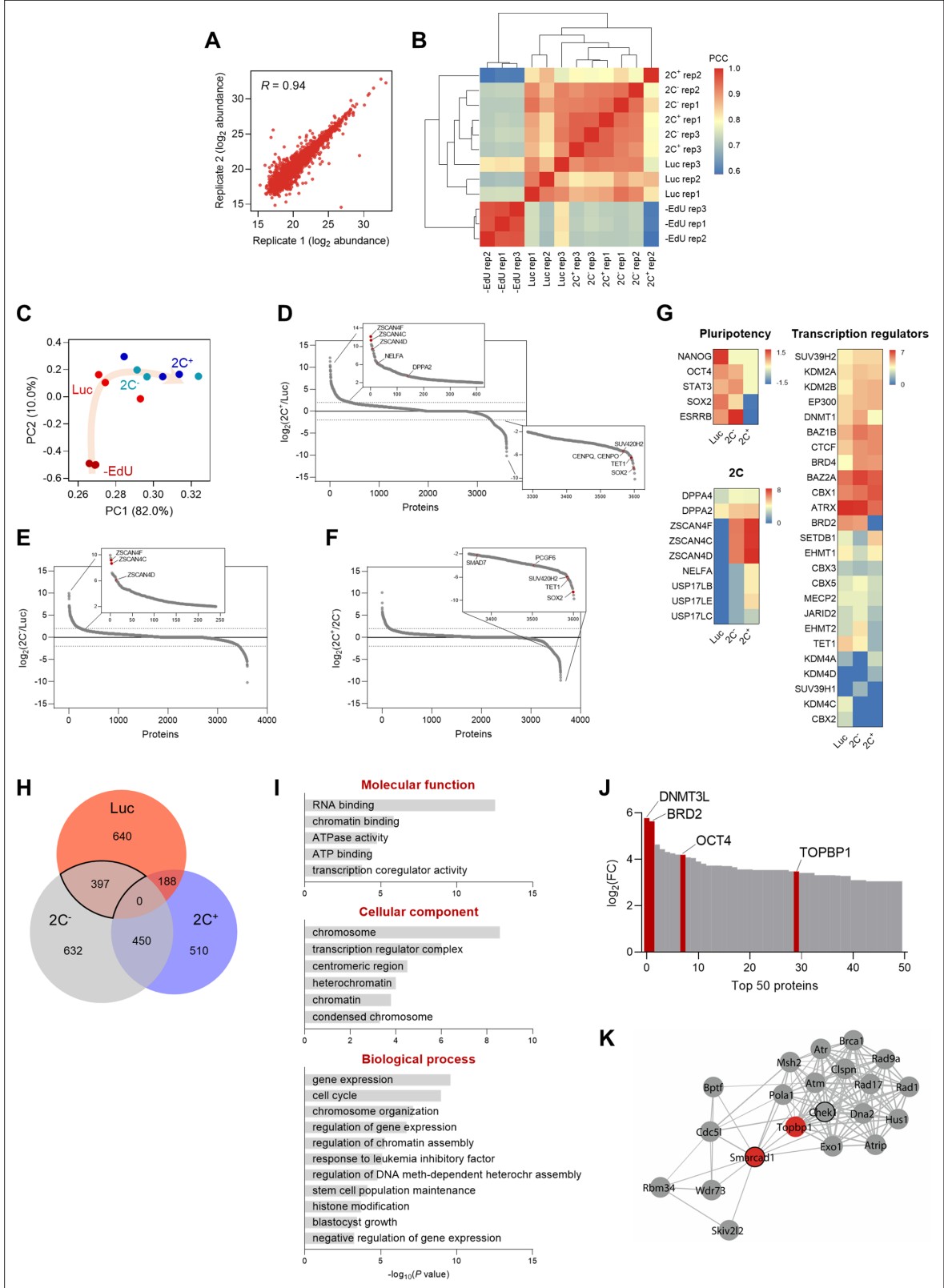

**Figure 2.** Chromatin-bound proteome profiling allows the identification of dynamic chromatome changes during 2C-like cell reprogramming. (**A**) Reproducibility between replicates of independent chromatome preparations. Correlation between replicate 1 and replicate 2 from the 2C⁻ condition is shown. R indicates Pearson's R. (**B**) Correlation matrix showing reproducibility among independent replicates of –EdU, Luc, 2C⁻ and 2C⁺ protein abundances. Hierarchical clustering analysis of the individual replicates is also shown. PCC, Pearson's correlation coefficient. (**C**) Principal component

*Figure 2 continued on next page*

*Figure 2 continued*

analysis (PCA) of the chromatome dataset. Each point corresponds to a single replicate. Beige arrow indicates the 2C-like reprogramming trajectory. (**D**) Protein enrichment analysis of the comparison between 2C$^+$ and Luc chromatomes. Red dots indicate known regulators of the 2C-like state, which were found enriched in the 2C$^+$ chromatome (upper panel), and novel factors that were found depleted from the 2C$^+$ chromatome (lower panel). Dashed lines indicate log$_2$ fold change (FC)±2. (**E**) Protein enrichment analysis of the comparison between 2C$^-$ and Luc chromatomes. Red dots indicate known regulators of the 2C-like state, which were found enriched in the 2C$^-$ chromatome. Dashed lines indicate log$_2$ FC±2. (**F**) Protein enrichment analysis of the comparison between 2C$^+$ and 2C$^-$ chromatomes. Red dots indicate novel factors that were found depleted from the 2C$^+$ chromatome. Dashed lines indicate log$_2$ FC±2. (**G**) Heat map representation of the chromatin-bound abundance of pluripotency transcription factors, 2-cell-specific factors, and transcriptional regulators in Luc, 2C$^-$ and 2C$^+$ cells. Data are presented as log$_2$ FC of PSM values to –EdU. PSM, peptide spectrum match. (**H**) Venn diagram indicating the overlap between the identified proteins enriched in Luc, 2C$^-$ and 2C$^+$ chromatomes after SAINT analysis. Specifically, an average enrichment value was computed from the respective pairwise comparisons (i.e. Luc vs 2C$^-$; Luc vs 2C$^+$; 2C$^-$ vs Luc; 2C$^-$ vs 2C$^+$), and proteins were selected on the basis of a minimum of FC≥2. The solid line highlights the 397 proteins enriched in the Luc and 2C$^-$ chromatomes that were not enriched in the 2C$^+$ chromatome. (**I**) Gene ontology analysis of the commonly enriched chromatin-bound proteins identified in (**H**). (**J**) Distribution of the top 50 chromatin-bound proteins identified in the Luc and 2C$^-$ chromatomes ranked by log$_2$ FC. (**K**) Functional protein network of TOPBP1 interactors. TOPBP1 and SMARCAD1 nodes are colored in red. The black node border indicates chromatin remodeling function. Network edges indicate the degree of confidence prediction of the interaction. Protein interaction data were retrieved from the STRING database (*Szklarczyk et al., 2017*).

The online version of this article includes the following source data and figure supplement(s) for figure 2:

**Figure supplement 1.** Chromatin proteomics of 2C-like cells.

**Figure supplement 1—source data 1.** Original membranes corresponding to *Figure 2—figure supplement 1A*.

**Figure supplement 1—source data 2.** Original membranes corresponding to *Figure 2—figure supplement 1A* with labels.

*De Iaco et al., 2017*; *Falco et al., 2007*), were identified among the top enriched factors in the 2C$^+$ chromatome (*Figure 2D, E, and G*). ZSCAN4 family members, such as ZSCAN4F and ZSCAN4C, were found associated with chromatin already in the 2C$^-$ chromatome (*Figure 2E*), supporting previous findings (*Rodriguez-Terrones et al., 2018*). However, we identified regulators of 2C-like cells such as TET1, the non-canonical Polycomb (PcG) Repressor Complex 1 (PRC1) member PCGF6, the TGF-β regulator SMAD7, and the heterochromatic H4K20me3 methyltransferase SUV420H2 depleted from the 2C$^+$ chromatome when compared to the 2C$^-$ (*Fu et al., 2019*; *Lu et al., 2014*; *Aloia et al., 2013*; *Figure 2D, F, and G*). The pluripotency transcription factors NANOG, OCT4, STAT3, SOX2, and ESRRB were, as expected, exclusively enriched in the 2C$^-$ and Luc chromatomes (*Figure 2G*). We also identified several transcriptional regulators and epigenetic enzymes differentially enriched in the 2C$^+$, 2C$^-$, and Luc chromatomes (*Figure 2G*). Interestingly, we identified marked differences in the enrichment for H3K9 histone methyltransferases SUV39H1 and SUV39H2 (*Figure 2G*). SUV39H2 gradually increased its abundance on chromatin as Luc cells converted to 2C$^-$ and, ultimately, to 2C$^+$ (*Figure 2G*). Contrarily, SUV39H1 was overall less abundantly chromatin-bound, although with a slight enrichment in the 2C$^-$ chromatome (*Figure 2G*). Altogether, these data indicate that ESC reprogramming toward 2C-like state correlates with a major reorganization of the chromatin-bound proteome.

We used the Significance Analysis of INTeractome (SAINT) algorithm (*Choi et al., 2011*) to further interrogate protein-chromatin interactions in the iPOTD datasets. To identify molecular drivers of chromocenter reorganization, we compared the enriched proteins in the 2C$^+$, 2C$^-$, and Luc chromatomes (*Figure 2H* and *Supplementary file 1*). We identified a total of 397 proteins shared by the 2C$^-$ and Luc chromatomes that were not enriched in the 2C$^+$ chromatome (*Figure 2H* and *Supplementary file 1*). We focused on analyzing this cluster since chromocenters are present in 2C$^-$ and Luc cells. This protein cluster included gene ontology (GO) terms associated with RNA and chromatin binding, active remodeling activity (e.g. ATPase activity), repressive chromatin (e.g. heterochromatin condensed chromosome, negative regulation of gene expression), and pluripotent stem cell identity (e.g. response to LIF, stem cell maintenance, blastocyst growth) (*Figure 2I*). To identify putative factors responsible for chromocenter reorganization, we ranked the commonly identified proteins included in 2C$^-$ and Luc chromatomes according to their fold change (*Figure 2J*). Notably, this protein cluster included known transcriptional regulators such as the DNA methyltransferase DNMT3L, the bromodomain-containing protein BRD2, the core pluripotency factor OCT4, and the DNA topoisomerase 2-binding protein 1, TOPBP1 (*Figure 2J*). We focused our attention on TOPBP1, which plays crucial roles in DNA replication and repair (*Yan and Michael, 2009*). Moreover, topoisomerases control genome structure and folding (*Wang, 2002*). We asked if the lack of topoisomerase activity could promote 2C$^+$ cell induction. Thus, we treated ESCs with camptothecin (CPT) and ICRF-193, inhibitors of DNA topoisomerases

I and II, respectively (*Pommier, 2006*; *Downes et al., 1994*). These compounds can indirectly recruit TOPBP1 to manage DNA repair following the inhibition of topoisomerase I and topoisomerase II activities. Inhibition of topoisomerase II alone increased the number of 2C⁺ cells 1.5-fold (*Figure 3— figure supplement 1A*) and triggered a prominent cell cycle arrest in the G2/M phase (*Downes et al., 1994*; *Robinson et al., 2007*; *Figure 3—figure supplement 1B*). Simultaneous inhibition of topoisomerases I and II resulted in an enhanced effect, leading to a 2.4-fold increase in the fraction of 2C⁺ cells (*Figure 3—figure supplement 1A and C*). These results motivated us to further investigate TOPBP1 network. TOPBP1 has been shown to interact with chromatin remodelers such as the SWI/SNF-like remodeler SMARCAD1 in yeast and human cells (*Bantele et al., 2017*; *Liu et al., 2004*; *Figure 2K*). We then investigated TOPBP1 and SMARCAD1 as potential candidate factors controlling the remodeling of chromocenters.

## SMARCAD1 and TOPBP1 associate with H3K9me3 in ESCs and can maintain heterochromatin foci

The results of the iPOTD revealed TOPBP1 as a potential regulator of chromocenter reorganization. SMARCAD1 has been shown to interact with TOPBP1 in yeast and human cells (*Bantele et al., 2017*; *Figure 2K*). Interestingly, SMARCAD1, an SWI/SNF-like chromatin remodeler, is known to promote heterochromatin maintenance during DNA replication in terminally differentiated cells and silencing of endogenous retroviruses (ERVs) in ESCs (*Sachs et al., 2019*; *Rowbotham et al., 2011*). Nonetheless, it is not known whether SMARCAD1 plays a role in 2C-like fate transition and early embryo development.

We, therefore, decided to investigate SMARCAD1 and TOPBP1 in 2C⁺ cells undergoing the transition to ESC-like cells, where chromocenters are formed de novo. We found that SMARCAD1 co-localized with H3K9me3 in heterochromatin foci of chromocenters in both ESCs and 2C⁻ cells (*Figure 3A–C*). In contrast, the expression of SMARCAD1 decreased in 2C⁺ cells, where foci were much reduced in number (*Figure 3B, C* and *Figure 3—figure supplement 1D*). We then asked whether SMARCAD1 depletion would increase the fraction of 2C⁺ cells. We depleted *Smarcad1* using two independent single guide RNAs (sgRNAs), as confirmed comparing to a control sgRNA targeting luciferase (*Figure 3—figure supplement 1E*). SMARCAD1 depletion resulted in no major impact in the 2C⁺ conversion either in the endogenous fluctuation or in Dux-induced cells when inspected at the steady state (*Figure 3D*). We then investigated *Smarcad1*-depleted cells 24 hr, 48 hr, and 72 hr after the 2C⁺ exit. Control KO cells (sgLuc) followed comparable exit kinetics compared with non-transfected (NT) 2C⁺ cells (*Figure 3—figure supplement 1F*). However, SMARCAD1 depletion resulted into a tendency to increased percentage of 2C⁺ cells at all time points after the exit (*Figure 3E*). Accordingly, the nuclear distribution of SMARCAD1 during exit from the 2C-like state changed. We first observed a diminution in SMARCAD1 signal as ESCs started to express the MERVL reporter, attaining severe reduction of SMARCAD1 in 2C⁺ at the 24 hr time point (*Figure 3G and H*). SMARCAD1 nuclear signal was then gradually recovered in the heterochromatin foci as 2C⁺ cells were converted in ESC-like cells up to the 72 hr from the exit, indicating reversibility of foci formation (*Figure 3G and H*). Surprisingly, the fraction of cells that repressed retroelements within 24 hr from the 2C⁺ exit (ESC-like at 24 hr) already showed SMARCAD1 enriched foci (*Figure 3G*). Altogether, these results suggest that SMARCAD1 was severely reduced from chromatin as ESCs progress to the 2C-like state and, later, SMARCAD1 nuclear distribution was reverted during the 2C⁺ exit.

We then used published single-cell RNA-seq data (*Deng et al., 2014*) and found *Smarcad1* expression starting at the 2-cell stage, but increasing at the 4-cell stage embryo, which is the time when chromocenters compact during mouse embryo development (*Figure 3I*). Notably, *Topbp1* showed a similar expression profile during preimplantation development (*Figure 3I*). Similar to what observed for *Smarcad1*-depleted cells, *Topbp1*-depleted cells showed a tendency to increased percentage of 2C⁺ cells at 24 hr, 48 hr, and 72 hr after 2C⁺ exit (*Figure 3F*). To further confirm these results and to investigate the role of TOPBP1 in the regulation of heterochromatin foci, we generated knocked-down ESC clones carrying shSmarcad1 or shTopbp1 (*Figure 3—figure supplement 2A*). We observed that the number of foci decreased and their area become larger after either knocking down *Smarcad1* or *Topbp1*, with respect to scramble controls (*Figure 3—figure supplement 2B and C*). Moreover, larger and fewer chromatin foci were visible in 2C⁺ cells when compared to ESCs and 2C⁻ cells (*Figure 3— figure supplement 2D*). We confirmed these results investigating *Topbp1*-depleted cells 24 hr, 48 hr,

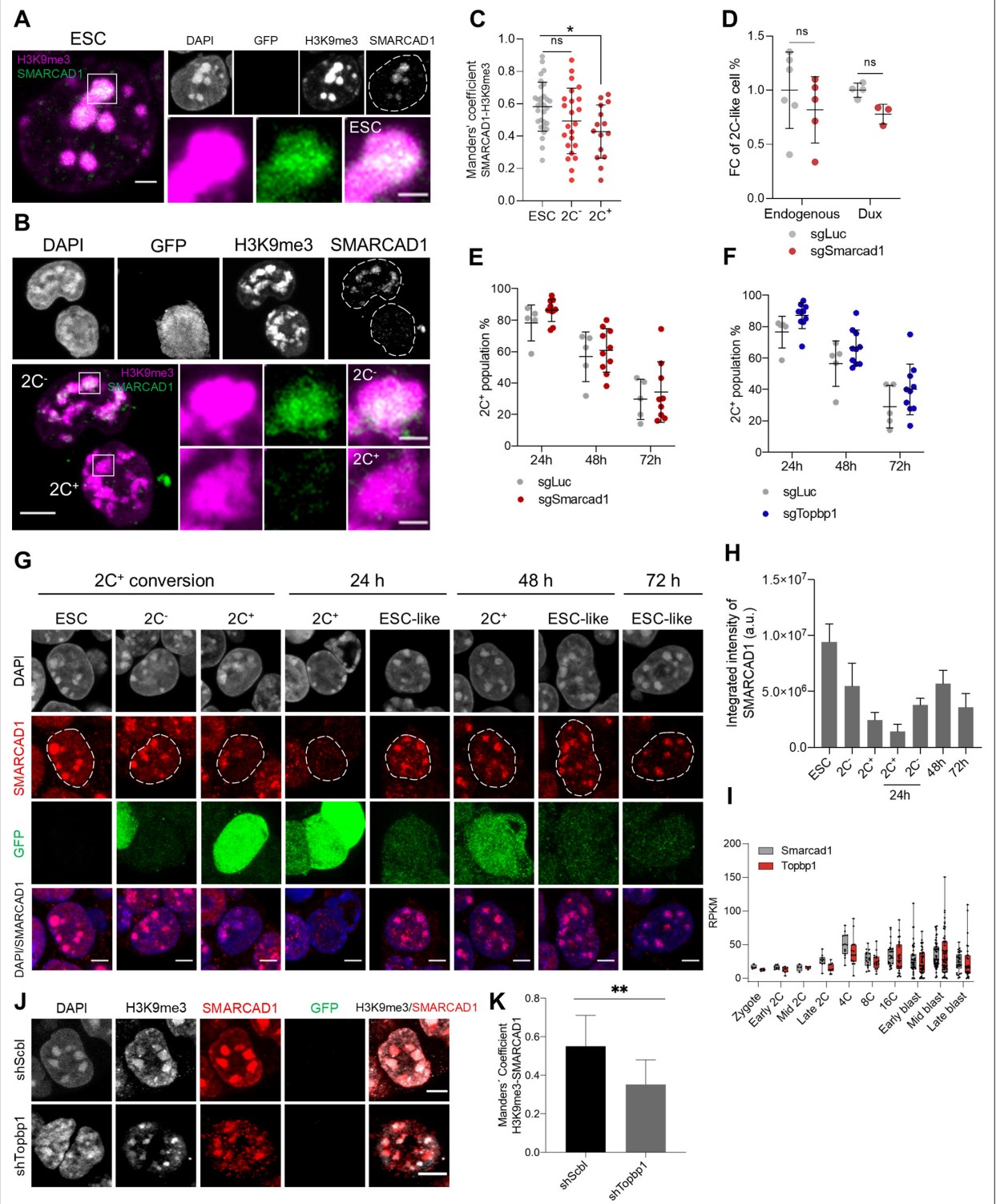

**Figure 3.** SMARCAD1 associates with H3 lysine 9 trimethylation (H3K9me3) in embryonic stem cells (ESCs) and its nuclear localization is reduced in the 2C-like state. (**A**) Representative immunofluorescence images of H3K9me3 and SMARCAD1 in ESCs. Dashed lines indicate nuclei contour. Scale bar, 2 μm. Zoomed images of H3K9me3 and SMARCAD1 foci are shown for comparisons. Scale bar, 1 μm. (**B**) Representative immunofluorescence images of H3K9me3 and SMARCAD1 in 2C- and 2C+ cells. Dashed lines indicate nuclei contour. Scale bar, 5 μm. Zoomed images of H3K9me3 and SMARCAD1 foci are shown for comparisons. Scale bar, 1 μm. (**C**) Co-localization analysis showing Manders' coefficient between SMARCAD1 and H3K9me3 in ESCs, 2C- and 2C+ cells. Data are presented as scatter dot plots with line at mean ± SD from ESC (n=30), 2C- (n=23), 2C+ (n=15) SMARCAD1-H3K9me3 foci. p>0.05[ns], p=0.0124* by one-way ANOVA (Dunnett's multiple comparisons test). (**D**) Impact of targeting *Smarcad1* (sgSmarcad1) on the endogenous

*Figure 3 continued on next page*

*Figure 3 continued*

fluctuation and the Dux-induced 2C-like conversion. Data are presented as scatter dot plots with line at mean ± SD (n≥3 independent CRISPR-Cas9 KO rounds). p=0.4286[ns], p=0.0571[ns] by Mann-Whitney test. (**E**) Impact of targeting *Smarcad1* (sgSmarcad1) on the 2C-like cell percentage during the 2C[+] exit (24 hr, 48 hr, and 72 hr). Data are presented as scatter dot plots with line at mean ± SD (n=5 independent CRISPR-Cas9 KO rounds). Individual points indicate scores of technical replicates. p=0.1174[ns] at 24 hr, p=0.6158[ns] at 48 hr, p=0.6441[ns] at 72 hr by multiple *t*-test. (**F**) Impact of targeting *Topbp1* (sgTopbp1) on the 2C-like cell percentage during the 2C[+] exit (24 hr, 48 hr, and 72 hr). Data are presented as scatter dot plots with line at mean ± SD (n=5 independent CRISPR-Cas9 KO rounds). Individual points indicate scores of technical replicates. p=0.0503[ns] at 24 hr, p=0.1589[ns] at 48 hr, p=0.2166[ns] at 72 hr by multiple *t*-test. (**G**) Representative immunofluorescence images of SMARCAD1 and the 2C::EGFP reporter along the ESCs to 2C[+] reprogramming and during the 2C[+] exit (24 hr, 48 hr, and 72 hr). Dashed lines indicate nuclei contour. Scale bar, 4 μm. (**H**) SMARCAD1 integrated intensity analysis along the conversion of ESCs into 2C[+] cells and during the 2C[+] exit (24 hr, 48 hr, and 72 hr). Data are presented as mean ± SD. (**I**) Single-cell RNA-seq (scRNA-seq) expression profile of *Smarcad1* and *Topbp1* in preimplantation mouse embryos. Data are presented as min-max boxplots with line at median. Each dot represents a single cell. scRNA-seq data was obtained from ***Deng et al., 2014***. RPKM, reads per kilobase of transcript per million mapped reads. (**J**) Representative immunofluorescence images of H3K9me3, SMARCAD1, and the 2C::EGFP reporter in *Topbp1* knockdown (shTopbp1) and control scramble (shScbl) cells. Scale bar, 5 μm. (**K**) Co-localization analysis showing Manders' coefficient between H3K9me3 and SMARCAD1 in *Topbp1* knockdown (shTopbp1) and control scramble (shScbl) cells. Data are presented as mean ± SD (n=2 independent cultures). p=0.0066** by unpaired two-tailed Student's *t*-test.

The online version of this article includes the following source data and figure supplement(s) for figure 3:

**Figure supplement 1.** Pharmacological and genetic perturbations in 2C-like cells.

**Figure supplement 1—source data 1.** Original membranes corresponding to *Figure 3—figure supplement 1D*.

**Figure supplement 1—source data 2.** Original membranes corresponding to *Figure 3—figure supplement 1D* with labels.

**Figure supplement 2.** H3 lysine 9 trimethylation (H3K9me3) foci analysis in *Smarcad1* and *Topbp1* knockdown embryonic stem cells (ESCs), 2C[-] and 2C[+] cells.

and 72 hr after the 2C[+] exit. We observed a decreased number of foci and their larger area at all time points analyzed after the exit (*Figure 3—figure supplement 2E–G*). These data suggest that heterochromatin foci are maintained in ESCs by SMARCAD1 and TOPBP1 and the depletion of both of these two proteins leads to a remodeling of the H3K9me3 foci.

Next, we asked about the functional interaction of SMARCAD1 and TOPBP1 and thus we evaluated the localization of SMARCAD1 after knocking down *Topbp1*. We found a significant reduction of SMARCAD1 co-localization with H3K9me3 in heterochromatin foci in *Topbp1*-depleted cells (*Figure 3J and K*), suggesting that SMARCAD1 and TOPBP1 might work as complex in the maintenance of heterochromatin foci.

## SMARCAD1 and TOPBP1 are necessary for early embryo development

Collectively, our findings suggested that both SMARCAD1 and TOPBP1 could be potential regulators of H3K9me3 heterochromatin in the 2C[+] transition. With this in mind, we aimed at investigating their function in preimplantation embryos. We injected zygote-stage (E0.5) embryos with morpholino antisense oligos (MOs) targeting *Smarcad1* or *Topbp1* along with a scrambled control morpholino (Ctrl MO) (*Figure 4A*, *Figure 4—figure supplement 1A*). As expected from MO, which acts by blocking translation, SMARCAD1 was degraded from the 2-cell stage, and a reduction in its levels was observed up to the 8-cell stage, in *Smarcad1* MO-injected embryos (*Figure 4—figure supplement 1B and C*). We could not image the degradation of TOPBP1 since available anti-TOPBP1 antibodies provide unspecific signal in immunofluorescence experiments. It is noteworthy that SMARCAD1 localizes exclusively in the nucleus of preimplantation embryos (*Figure 4—figure supplement 1B*). We observed that embryos developed slower than normal when *Smarcad1* was silenced (*Figure 4A and B*). Indeed, they did not show the formation nor expansion of a blastocoel cavity at the early blastocyst stage, indicating a severe developmental delay (*Figure 4A and B*). Notably, 68% of the embryos deficient for *Smarcad1* arrested and did not develop until the late blastocyst stage (*Figure 4A and B*). In the case of *Topbp1* silencing, we observed an even more severe phenotype. All the embryos, 100% of the *Topbp1* MO-injected ones, did not develop and arrest at 4-cell stage (*Figure 4A and B*).

Since we observed that both SMARCAD1 and TOPBP1 were necessary for embryo developmental progression, we decided next to image H3K9me3 upon depletion of SMARCAD1 or of TOPBP1 (*Figure 4—figure supplement 1A*). H3K9me3 signal was significantly reduced in the embryos injected with *Smarcad1* MO already at the 8-cell stage (E2.5), almost 1 day earlier than early blastocyst (E3.5), when the developmental delay was morphologically visible (*Figure 4C, D*, *Figure 4—figure*

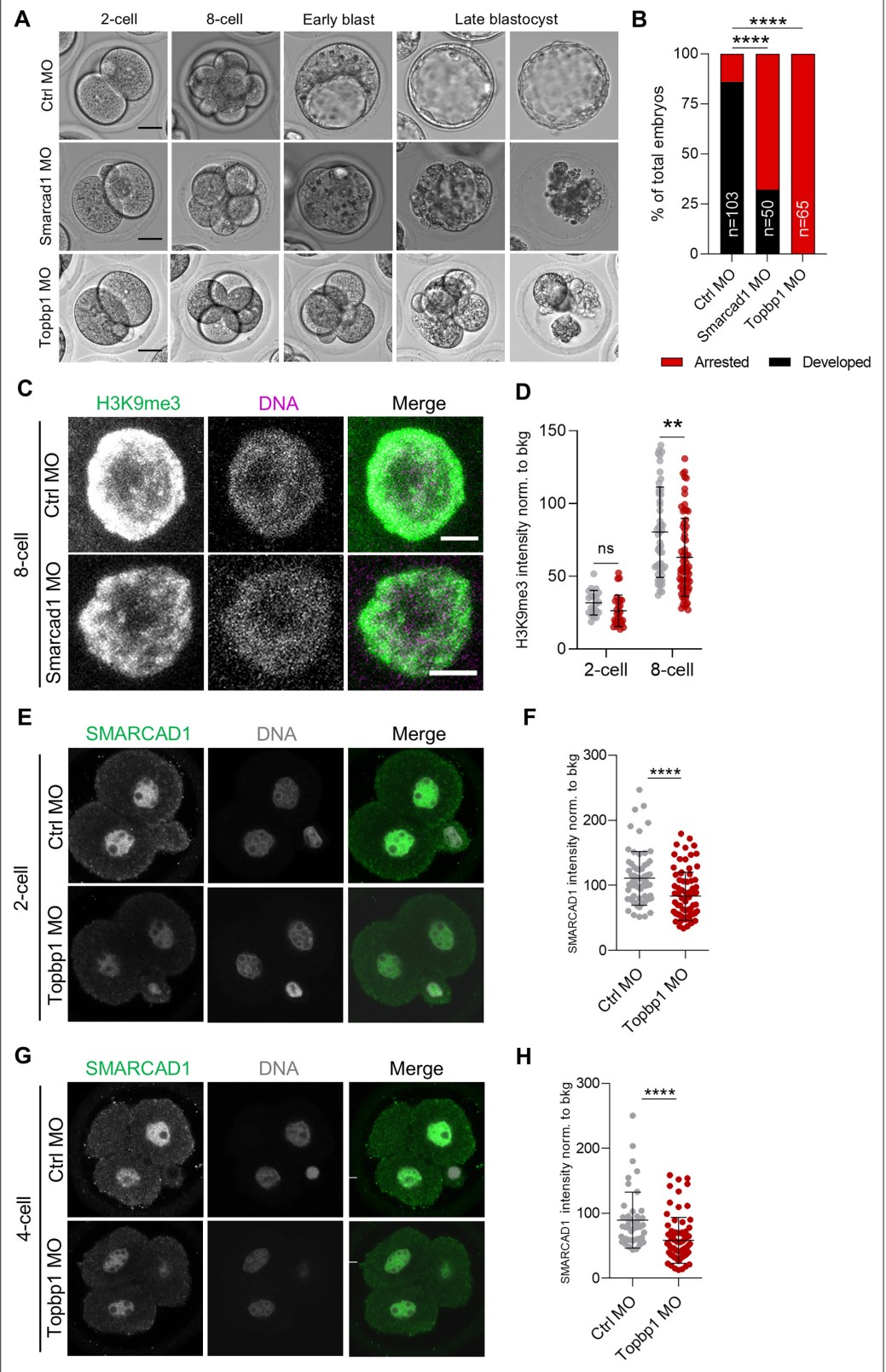

**Figure 4.** SMARCAD1 and TOPBP1 downregulation impairs embryo development. (**A**) Representative embryos from control (Ctrl), *Smarcad1* and *Topbp1* morpholino-injected (MO) groups from 2-cell (E1.5) to late blastocyst stage (E5.5). Scale bar, 20 μm. (**B**) Quantification of the percentage of arrested or fully developed embryos at late blastocyst stage (E4.5). p<0.0001**** by Fisher's exact test (Ctrl MO=103 embryos, *Smarcad1* MO=50 embryos,

*Figure 4 continued on next page*

*Figure 4 continued*

*Topbp1* MO=65 embryos). (**C**) Representative immunofluorescence images of H3 lysine 9 trimethylation (H3K9me3) in Ctrl and *Smarcad1* MO embryos at 8-cell stage (E2.5) embryos. Representative blastomere nuclei are shown. Scale bar, 5 µm. (**D**) Quantification of H3K9me3 mean fluorescence intensity in control (Ctrl, gray dots) and *Smarcad1* MO (red dots) embryos at 2-cell (E1.5) and 8-cell stage (E2.5). Data are presented as scatter dot plots with line at mean ± SD (2-cell: Ctrl MO=12 embryos, *Smarcad1* MO=15 embryos; 8-cell: Ctrl MO=16 embryos, *Smarcad1* MO=20 embryos). H3K9me3 signal was normalized to the average background signal. $p=0.0618^{ns}$ and $p=0.0016^{**}$ by unpaired two-tailed Student's *t*-test. (**E**) Representative immunofluorescence images of SMARCAD1 in Ctrl and *Topbp1* MO embryos at 2-cell stage (E1.5) embryos. Representative blastomere nuclei are shown. Scale bar, 10 µm. (**F**) Quantification of SMARCAD1 mean fluorescence intensity in Ctrl and *Topbp1* MO embryos at 2-cell (E1.5). Data are presented as scatter dot plots with line at mean ± SD (Ctrl MO=38 embryos, *Topbp1* MO=44 embryos). SMARCAD1 signal was normalized to the average background signal. $p<0.0001^{****}$ by unpaired two-tailed Student's *t*-test. (**G**) Representative immunofluorescence images of SMARCAD1 in Ctrl and *Topbp1* MO embryos at 4-cell stage (E2.0) embryos. Representative blastomere nuclei are shown. Scale bar, 10 µm. (**H**) Quantification of SMARCAD1 mean fluorescence intensity in Ctrl and *Topbp1* MO embryos arrested at 4-cell. Data are presented as scatter dot plots with line at mean ± SD (Ctrl MO=20 embryos, *Topbp1* MO=31 embryos). SMARCAD1 signal was normalized to the average background signal. $p<0.0001^{****}$ by unpaired two-tailed Student's *t*-test.

The online version of this article includes the following figure supplement(s) for figure 4:

**Figure supplement 1.** SMARCAD1 downregulation in mouse embryos.

---

*supplement 1A*). In *Topbp1* MO embryos, we did not observe decreased intensity of the H3K9me3 signal since the developmental arrest was present already at 4-cell stage and variation in this histone mark might be clearly measurable only starting from morula stage (*Figure 4—figure supplement 1A, D, and E*). On the other hand, we analyzed HP1β, a major component of constitutive heterochromatin which binds to both DNA and to H3K9me3 (*Lachner et al., 2001*; *Zhao et al., 2000*). We observed a major remodeling of heterochromatin in both 2-cell and 4-cell Topbp1 MO arrested embryos, as indicated by the spreading and increased signal of HP1β (*Figure 4—figure supplement 1F–I*).

Finally, given that we observed SMARCAD1 reduction in heterochromatin foci in *Topbp1*-depleted cells (*Figure 3J and K*), we investigated SMARCAD1 level in *Topbp1* MO in 2-cell and 4-cell arrested embryos. We observed a severe reduction of SMARCAD1 that was even more pronounced when analyzing the pool of 2-cell arrested embryos (*Figure 4E–H* and *Figure 4—figure supplement 1J-L*).

Collectively, these results confirm the functional interaction between SMARCAD1 and TOPBP1, showing that *Smarcad1* or *Topbp1* knockdown impairs mouse embryo development and that their role in the maintenance of H3K9me3 heterochromatin foci might contribute to the developmental arrest. Overall, our results suggest that both SMARCAD1 and TOPBP1 contribute to proper early embryo development.

## Discussion

Heterochromatin formation during early embryogenesis is a fundamental aspect of development (*Probst and Almouzni, 2011*). Here, we have reported that the transition from the 2C-like to the pluripotent state is a robust in vitro model system to study heterochromatin foci establishment and their reorganization in early embryo development. During the 2C-like to pluripotency transition, we found that heterochromatin foci are re-formed along with the DNA compaction of the chromatin fibers. Unlike previous reports that focused exclusively on transcriptional changes (*Fu et al., 2019*; *Rodriguez-Terrones et al., 2018*; *Fu et al., 2020*), our study exploited chromatin proteomics by genome capture to unravel an additional layer of information and complexity in the 2C-like system. Thus, we provided a detailed characterization of the stepwise chromatome dynamics occurring during the 2C-like state transition. Remarkably, we identified the chromatin remodeler factor SMARCAD1 and TOPBP1, a binding protein interacting with topoisomerase activity, to contribute to embryo development. Depletion of SMARCAD1 or TOPBP1 in preimplantation embryos led to severe developmental arrest and to a substantial remodeling of H3K9me3 heterochromatin foci. These findings have important implications because the establishment and maintenance of heterochromatin foci during embryo development is a key step in the embryonic totipotent program of the 2-cell stage toward pluripotency (*Burton and Torres-Padilla, 2014*; *Zernicka-Goetz et al., 2009*).

ERVs are transposable elements flanked by long terminal direct repeats (*Rodriguez-Terrones and Torres-Padilla, 2018*; *Friedli and Trono, 2015*). Tight control of ERVs and their transposable activity is essential for genome integrity and plays an important role in early development and pluripotency (*Rodriguez-Terrones and Torres-Padilla, 2018*; *Friedli and Trono, 2015*). H3K9me3 has been associated with retrotransposons through the KRAB-associated protein 1, KAP1 (*Rowe et al., 2010*). KAP1 led to the silencing of ERVs in ESCs by inducing H3K9me3 heterochromatin formation via the recruitment of the H3K9 histone methyltransferase SETDB1 (*Rowe et al., 2010*; *Matsui et al., 2010*; *Wolf and Goff, 2007*). SMARCAD1 was discovered recently to directly interact with KAP1 and therefore be an important regulator of the KAP1-SETDB1 silencing complex in ESCs (*Sachs et al., 2019*; *Ding et al., 2018*). SMARCAD1 is also a key factor for ERV silencing in ESCs (*Sachs et al., 2019*), where it remodels nucleosomes (*Navarro et al., 2020*). Of note, although SMARCAD1 is highly expressed in ESCs, its depletion does not affect pluripotency (*Sachs et al., 2019*; *Navarro et al., 2020*; *Xiao et al., 2017*).

SMARCAD1 has been described in ESCs, yet its function in 2C-like cells has not been explored. Our observation that SMARCAD1 enriches in H3K9me3 heterochromatin foci during the transition from the 2C-like state to pluripotency and that it contributes to early mouse embryo development is aligned with the observations previously reported in ESCs. It will be interesting in the future to study whether SMARCAD1 can tether the KAP1-SETDB1 to directly induce the formation of H3K9me3 heterochromatin foci at the exit of the 2-cell stage in the embryos. Recently, the H3K9 histone methyltransferase SUV39H2 has been reported to catalyze de novo H3K9me3 in the paternal pronucleus after fertilization (*Burton et al., 2020*). Yet, *Suv39h2* downregulation in zygote-stage embryos did not translate on appreciable changes in H3K9me3 levels on the maternal chromatin. This opens up the possibility that different methyltransferases, and their regulators like SMARCAD1, could be responsible for H3K9me3 acquisition in this early developmental stage.

Topoisomerases likely cooperate with the chromatin remodeling factor SMARCAD1 in yeast (*Bantele et al., 2017*). This is in line with our observations that SMARCAD1 is reduced in heterochromatin foci in both *Topbp1*-depleted cells and *Topbp1*-depleted embryos. However, it remains unclear whether SMARCAD1 functions independently or as a part of a large remodeling complex.

We showed that topoisomerase inhibition led to an increase in the fraction of 2C-like cells and cell cycle arrest in the G2/M phase. Moreover, the knockdown of TOPBP1 leads to a severe developmental arrest. Thus, it is also tempting to speculate that cell cycle progression, especially since we observed that 2C$^+$ cells might be arrested in the G2/M phase, has a role in regulating SMARCAD1 recruitment and/or function on chromatin during the 2C$^+$ exit. Additionally, it should be noted that DNA damage response (DDR) and p53 have been reported to activate Dux in vitro, and thus, DDR and associated factors may contribute to the increased percentage of 2C$^+$ cells observed upon topoisomerase inhibition (*Atashpaz et al., 2020*; *Grow et al., 2021*). In the in vivo scenario, this prolonged G2/M phase might be necessary to rewire specific epigenetic modifications in the 2-cell blastomeres to allow heterochromatin formation or control DNA repair. TOPBP1, being a DNA topoisomerase 2-binding protein and involved in DNA repair (*Bagge et al., 2021*), might have a role in this process. This is a key step before the blastomeres can embark into the correct developmental process, as proposed for early *Drosophila* embryos (*Seller et al., 2019*).

By using chromatin proteomics, we have provided additional data that will help to elucidate the molecular intricacies of the 2C-like state and early mammalian development. In the current study, we focused on heterochromatin establishment and we identified SMARCAD1 and TOPBP1, which both interact with H3K9me3. SMARCAD1 might act in the complex as the remodeler factor that, by regulating methyltransferases, can facilitate H3K9me3 deposition at the exit of the totipotent 2-cell stage when heterochromatin is established de novo. Although we could not collect robust data on the alteration of the 2C program, we have indication of its prolonged activity when either SMARCAD1 or TOPBP1 are knocked down, in line with a role in regulating early development in the maintenance of heterochromatin and the regulation of the 2C program.

# Materials and methods

## Cell lines and culture conditions

E14Tg2a mouse ESCs were cultured in gelatinized plates in high glucose DMEM supplemented with 15% FBS (Sigma), GlutaMAX, sodium pyruvate, non-essential amino acids, penicillin/streptomycin, 100 µM 2-mercaptoethanol, 1000 U/ml mouse leukemia inhibitory factor (Millipore), 1 µM PD0325901, and 3 µM CHIR99021. After viral infection, ESCs were selected and maintained with ES medium containing the appropriate combination of selection drugs (250 µg/ml Geneticin [G418, Life Technologies], 0.5 µg/ml Puromycin [Life Technologies]). ESCs were treated with 2 µg/ml doxycycline (D9891, Sigma) for 24 hr to induce Dux expression. The Dux overexpression system was benchmarked according to previously reported features. Dux overexpression resulted in the loss of DAPI-dense chromocenters and the loss of the pluripotency transcription factor OCT4 (*Figure 1—figure supplement 1E*; *Macfarlan et al., 2012*; *Ishiuchi et al., 2015*), upregulation of specific genes of the 2-cell transcriptional program such as endogenous Dux, MERVL, and major satellites (MajSat) (*Figure 1—figure supplement 1F*; *Macfarlan et al., 2012*; *Ishiuchi et al., 2015*; *Hendrickson et al., 2017*; *De Iaco et al., 2017*; *Whiddon et al., 2017*), and accumulation in the G2/M cell cycle phase (*Figure 1—figure supplement 1G*), with a reduced S phase consistent in several clonal lines (*Figure 1—figure supplement 1H*; *Eckersley-Maslin et al., 2016*).

## Lentivirus production and ESC infection

Lentiviral particles were produced following the RNA interference Consortium (TRC) instructions for viral production and cell infection (http://www.broadinstitute.org/rnai/public/). HEK293T cells were co-transfected with the lentiviral plasmid of interest (pCW57.1-Luciferase or pCW57.1-mDux-CA) and the viral packing vectors (pCMV-ΔR8.9 and pCMV-VSV-G) using the CalPhos mammalian transfection kit (631312, Clontech). pCW57.1-Luciferase and pCW57.1-mDux-CA were a gift from Stephen Tapscott (Addgene plasmids #99283 and #99284). Short hairpins targeting *Smarcad1* (shSmarcad1), *Topbp1* (shTopbp1 #1 and shTopbp1 #2), and a scramble control sequence (shScbl) were cloned into the pLKO.1-Hygro lentiviral vector (Addgene plasmid #24150). The oligos used for cloning shRNAs are listed in *Table 1*. The lentiviral-containing medium was harvested from HEK293T cells at 48 hr and 72 hr after transfection, filtered and used for ESC infection. Two days after the last round of infection, ESCs were selected with the indicated concentration of the selection drug (see Cell lines and culture conditions).

## Fluorescence-activated cell sorting

Quantification of GFP positive cells and cell cycle analysis was performed with a LSR II Analyzer (BD Biosciences). For cell sorting, an Influx Cell Sorter (BD Biosciences), was used to sort the specified populations in each experiment.

## Cell cycle analysis by flow cytometry

For cell cycle analysis of live cells, $5×10^4$ ESCs were plated per well in gelatin-coated six-well plates 1 day before starting the experiment. At the moment of the assay, ESCs were trypsinized, collected, and washed with PBS before incubation with ES medium supplemented with 10 µg/ml Hoechst 33342 (H1399, Thermo Fisher) for 30 min at 37°C. Propidium iodide (1 µg/ml; P4864, Sigma) was added to stain dead cells. All flow cytometry data were processed and analyzed with FlowJo (v10).

**Table 1.** List of top oligos used for cloning shRNAs.

| shRNA name | Top oligo (5' to 3') |
| --- | --- |
| shScbl | GTCACGATAAGACAATGAT |
| sh*Smarcad1* | CCTCCCTTCTAAACCAAAGTT |
| sh*Topbp1* #1 | CCTGAATTTGAATCACTGGTT |
| sh*Topbp1* #2 | GCTCTTAGAAACTGCGAGAAT |

## Inhibition of DNA topoisomerases

To inhibit DNA topoisomerases, ESCs were treated with 500 nM of the topoisomerase I inhibitor CPT (ab120115, Abcam) and/or with 5 µM of the topoisomerase II inhibitor ICRF-193 (I4659, Sigma) for 12 hr.

## Immunostaining, image processing, and quantification

### Immunofluorescence staining of ESCs

ESCs were plated at a concentration of 56,000 cells/cm$^2$ in gelatin-coated borosilicate glass bottom Nunc Lab-Tek (155411, Thermo Fisher) or µ-Slide (80827, Ibidi) eight-well chambers. Cells were fixed with 4% paraformaldehyde (PFA) for 10 min and were then washed three times with PBS. Cells were permeabilized and blocked (10% GS, 2.5% BSA, 0.4% Triton X-100) for 30 min at room temperature (RT). Incubation with the corresponding primary antibodies at the indicated dilutions lasted 3 hr at 37°C. Cells were then washed and incubated with Alexa Fluor (Molecular Probes, Invitrogen) secondary antibodies for 1 hr at RT. For H3K9me3 and SMARCAD1 co-staining, cells were washed three times with PBS after secondary antibody incubation. Then, cells were incubated with second primary antibody and the corresponding secondary antibody as indicated above. Finally, cells were washed three times with PBS containing DAPI for nuclear counterstain. Images were acquired on a Leica TCS SP5 confocal microscope equipped with a ×63 oil objective.

The following antibodies were used: chicken anti-GFP (1:500; ab13970, Abcam), mouse anti-Oct-3/4 (1:200; sc-5279, Santa Cruz), rabbit anti-histone H3K9me3 (1:500; ab8898, Abcam), mouse anti-SMARCAD1 (1:500; ab67548, Abcam), goat anti-chicken Alexa Fluor 488, goat anti-mouse Alexa Fluor 568, goat anti-rabbit Alexa Fluor 568, goat anti-mouse Alexa Fluor 647. All secondary antibodies were provided by Molecular Probes (Invitrogen).

### EdC incorporation and DNA labeling

To label DNA, a 14 hr incorporation pulse of EdC (T511307, Sigma) at 2.5 µM was performed in ESCs, in parallel to doxycycline treatment. Cells were plated in gelatin-coated borosilicate glass bottom chambers at a concentration of 56,000 cells/cm$^2$ in ES medium supplemented with EdC for 14 hr. At the end of EdC incorporation, ESCs were fixed with PFA 4% (43368, Thermo Fisher Alfa Aesar) and permeabilized with 0.4% Triton X-100. Click chemistry reaction was performed by incubating cells for 30 min at RT in click chemistry buffer: 100 mM HEPES pH 8.2, 50 mM amino guanidine (396494, Sigma), 25 mM ascorbic acid (A92902, Sigma), 1 mM CuSO$_4$, 2% glucose (G8270, Sigma), 0.1% Glox solution (0.5 mg/ml glucose oxidase, 40 mg/ml catalase [G2133 and C100, Sigma]) and 10 mM Alexa Fluor 647 Azide (A-10277, Thermo Fisher) (*Otterstrom et al., 2019*; *Zessin et al., 2012*; *Raulf, 2014*). After washing the samples three times with PBS, we directly proceeded to perform STORM imaging.

### STORM imaging

Stochastic Optical Reconstruction Microscopy (STORM) imaging was performed on an N-STORM 4.0 microscope (Nikon) equipped with a CFI HP Apochromat TIRF ×100 1.49 oil objective and a iXon Ultra 897 camera (Andor) with a pixel size of 16 µm. This objective/camera combination provides an effective pixel size of 160 nm. STORM images were acquired with 10 ms exposure time for 60,000 frames using highly inclined (HILO) illumination. An activator/reporter pair strategy was used with AF405 and AF647 fluorophores, respectively. Continuous imaging acquisition was performed with simultaneous 405 nm and 647 nm illumination. 647 nm laser was used at constant ~2 kW/cm$^2$ power density. 405 nm laser was used at low laser power and gradually increased during the imaging to enhance fluorophore reactivation and to maintain the density of localizations per frame constant. Before STORM imaging, we acquired conventional fluorescence images of GFP for each nucleus to discriminate between 2C$^-$ and 2C$^+$ cells. Imaging buffer composition for STORM imaging was 100 mM cysteamine MEA (30070, Sigma), 1% Glox solution, and 5% glucose (G8270, Sigma) in PBS.

STORM images were analyzed and rendered in Insight3 as previously described (*Bates et al., 2007*; *Rust et al., 2006*). Localizations were identified based on an intensity threshold and the intensity distribution of their corresponding point spread functions fit with a 2D Gaussian to determine the x-y positions of their centers with high accuracy (~20 nm).

## Voronoi tessellation analysis

For Voronoi tessellation analysis, we used the list of localization from STORM (*Andronov et al., 2016*; *Levet et al., 2015*) and then we used a previously developed custom-made MATLAB script (*Otterstrom et al., 2019*). X-y coordinates of the localizations were used to generate the Voronoi polygons. Local densities were defined as the inverse value of the area of each Voronoi polygon. For visualization, we color-coded each Voronoi polygon based on their area, from yellow for the smallest polygons (density>0.01 nm$^{-2}$), to blue for larger polygons (density<0.0001 nm$^{-2}$). Finally, the largest 0.5% of polygons were set to black. For each nucleus, we computed the mean Voronoi density (nm$^{-2}$) as a measure of global DNA compaction.

For the GFP intensity score, we quantified the GFP conventional images (488 nm channel) with lower intensities in order to assign a GFP intensity score to each nucleus. We summed the fluorescence intensity ADU counts inside each nucleus and divided it by the total number of pixels to obtain the average GFP intensity. Then, we used the distribution of GFP intensities from the different nuclei to normalize the values, obtaining a GFP intensity score ranging from 0 (less bright) to 1 (most bright). We then performed a cell-by-cell analysis of the relation between GFP intensity score and global chromatin compaction obtained from Voronoi tessellation analysis.

## Immunofluorescence of preimplantation embryos

Preimplantation embryos at E1.5 and E2.5 stages were fixed with 2% PFA for 10 min at RT, permeabilized (0.25% Triton X-100) for 10 min, and then blocked (3% BSA) for 1 hr at 37°C. Incubation with the corresponding primary antibodies at the indicated dilutions in 1% BSA lasted one overnight at 4°C. After washing, embryos were incubated with Alexa Fluor (Molecular Probes, Invitrogen) secondary antibodies diluted in 1% BSA for 1 hr at 37°C. Finally, embryos were washed and transferred to an imaging buffer containing DRAQ5 (1:500; 62251, Thermo Fisher) for DNA staining. Images were acquired on a Leica TCS SP8 STED3X confocal microscope equipped with a ×63 oil objective.

The following antibodies were used: rabbit anti-histone H3K9me3 (1:500; ab8898, Abcam), mouse anti-SMARCAD1 (1:250; ab67548, Abcam), rabbit anti-HP1β (1:200; ab10478, Abcam), goat anti-rabbit Alexa Fluor 488, and goat anti-mouse Alexa Fluor 488. All secondary antibodies were provided by Molecular Probes (Invitrogen).

## Image processing and quantification

Immunofluorescence images were processed and analyzed with the ImageJ software (https://imagej.net/download/). All immunofluorescence images were acquired with z-stacks. Z-stacks were projected using the *maximum intensity* z-projection type. For SMARCAD1 nuclear signal analysis, manual selection of nuclear area was performed and integrated intensity was measured. For SMARCAD1-H3K9me3 co-immunofluorescence images, a Gaussian blur filtering (σ=0.5) was applied to the SMARCAD1

**Table 2.** List of oligos used for quantitative real-time PCR (qRT-PCR).

| Gene name | Forward (5' to 3') | Reverse (5' to 3') |
|---|---|---|
| *Dux* | GGAGAAGAGATACCTGAGCTTCAA | AATCTGAGACCCCCATTCG |
| *MERVL* | CTCTACCCACTTGGACCATATGAC | GAGGCTCCAAACAGCATCTCTA |
| *MajSat* | GCACACTGAAGGACCTGGAATATG | GATTTCGTCATTTTTCAAGTCGTC |
| *Zscan4* | GAGATTCATGGAGAGTCTGACTGATGAGTG | GCTGTTGTTTCAAAAGCTTGATGACTTC |
| *Nelfa* | TGCTAGTGGACACAGTGTTCGA | TTGAAGCGTGTCCACTGGCC |
| *Zfp352* | CCAGGACCCTGCAATACACA | TACAGGTTGTCTCCTGTGTGC |
| *Eif1a-like* | AACAGGCGCAGAGGTAAAAA | CTTATATGGCACAGCCTCCT |
| *Smarcad1* | AAATTCAGCAAAGACACAGTGATT | CAGAAGGAAGGTCATGGGATT |
| *Topbp1* | GCGCCACCAGCAATGTG | TGTACAGGATACAGTTACGTCAGACATTA |
| *Gapdh* | TCAAGAAGGTGGTGAAGCAGG | ACCAGGAAATGAGCTTGACAAA |
| *β-actin* | GCTGTATTCCCCTCCATCGTG | CACGGTTGGCCTTAGGGTTCAG |

channel. Fluorescence intensities of H3K9me3 foci were analyzed using the *3D Object Counter* function (https://imagej.net/3D_Objects_Counter, ImageJ). Co-localization analysis was done using the *JACoP* plugin (https://imagej.net/JaCoP, ImageJ). Manders' coefficient was calculated with the *JACoP* plugin. Manders' coefficient was used as a co-localization indicator because of its independence of the intensity of the overlapping pixels. For the quantification of H3K9me3 and SMARCAD1 fluorescence intensities in preimplantation embryos, manual selection of the nuclear area was performed for each blastomere. Fluorescent signals were measured and then normalized by the average cytoplasmic signal (background) in each condition. For the normalization step, the fluorescence intensity of a squared shape of equal size was taken for each individual blastomere.

## RNA extraction and qRT-PCR

RNA was extracted from pelleted or sorted ESCs using the RNA isolation RNeasy Mini kit (QIAGEN), according to the manufacturer's protocol. RNA was reverse-transcribed with iScript cDNA Synthesis kit (Bio-Rad). Quantitative real-time PCRs (qRT-PCRs) were performed using LightCycler 480 SYBR Green I Master (Roche) in a LightCycler 480 (Roche) instrument, according to the manufacturer's recommendations. The oligos used are listed in *Table 2*. qRT-PCR data was normalized to *Gapdh* or *β-actin* expression. For each sample, we had at least a technical duplicate.

## Chromatin-bound proteome profiling by genome capture (iPOTD)

ESCs were plated at a concentration of 34,000 cells/cm$^2$ in gelatin-coated 150 mm dishes. Then, ESCs were pulsed for 24 hr with 0.1 µM 5-ethynyl-2'-deoxyuridine (EdU; T511285, Sigma), in parallel to doxycycline treatment. Sorted luciferase (±EdU), 2C$^-$+EdU, and 2C$^+$+EdU cells were fixed with 1% PFA, quenched with 0.125 mM glycine (pH 7), and harvested immediately after sorting. Of note, ~10$^7$ cells were sorted per replicate and condition. Cells were later processed as described previously to extract the chromatin-bound proteins (*Aranda et al., 2020*; *Aranda et al., 2019*).

## Mass spectrometry analysis

### Sample preparation

Eluted proteins were reduced with dithiothreitol (37°C, 60 min) and alkylated in the dark with iodoacetamide (25°C, 20 min) prior to sequential digestion with endoproteinase LysC (1:10 wt:wt, 37°C, overnight; 129-02541, Wako) and trypsin (1:10 wt:wt, 37°C, 8 hr) according to filter-aided sample preparation procedure (*Wiśniewski et al., 2009*). After digestion, the peptide mixtures were acidified with formic acid and desalted with a MicroSpin C18 column (The Nest Group, Inc) prior to LC-MS/MS (liquid chromatography-tandem mass spectrometry) analysis.

### Chromatographic and mass spectrometric analysis

Samples were analyzed using a LTQ-Orbitrap Fusion Lumos mass spectrometer (Thermo Fisher Scientific, San Jose, CA, USA) coupled to an EASY-nLC 1200 (Thermo Fisher Scientific [Proxeon], Odense, Denmark). Peptides were loaded directly onto the analytical column and were separated by reversed-phase chromatography using a 50 cm column with an inner diameter of 75 µm, packed with 2 µm C18 particles spectrometer (Thermo Scientific, San Jose, CA, USA).

Chromatographic gradients started at 95% buffer A and 5% buffer B with a flow rate of 300 nl/min for 5 min and gradually increased to 22% buffer B and 78% A in 79 min and then to 35% buffer B and 65% A in 11 min. After each analysis, the column was washed for 10 min with 10% buffer A and 90% buffer B. Buffer A was 0.1% formic acid in water and buffer B was 0.1% formic acid in acetonitrile.

The mass spectrometer was operated in positive ionization mode with nanospray voltage set at 1.9 kV and source temperature at 275°C. Ultramark 1621 was used for external calibration of the FT mass analyzer prior to the analyses, and an internal calibration was performed using the background polysiloxane ion signal at m/z 445.1200. The acquisition was performed in data-dependent acquisition (DDA) mode and full MS scans with 1 micro scans at a resolution of 120,000 were used over a mass range of m/z 350–1500 with detection in the Orbitrap mass analyzer. Auto gain control (AGC) was set to 1E5 and charge state filtering disqualifying singly charged peptides was activated. In each cycle of DDA analysis, following each survey scan, the most intense ions above a threshold ion count of 10,000 were selected for fragmentation. The number of selected precursor ions for fragmentation was determined by the 'Top Speed' acquisition algorithm and a dynamic exclusion of 60 s. Fragment

ion spectra were produced via high-energy collision dissociation (HCD) at normalized collision energy of 28% and they were acquired in the ion trap mass analyzer. AGC was set to 1E4, and an isolation window of 1.6 m/z and maximum injection time of 200 ms were used. All data were acquired with Xcalibur software.

Digested bovine serum albumin (P8108S, NEB) was analyzed between each sample to avoid sample carryover and to assure stability of the instrument and QCloud has been used to control instrument longitudinal performance during the project (*Chiva et al., 2018*).

## Data analysis

Acquired spectra were analyzed using the Proteome Discoverer software suite (v2.3, Thermo Fisher Scientific) and the Mascot search engine (*Perkins et al., 1999*) (v2.6, Matrix Science). The data were searched against a Swiss-Prot mouse database (as in October 2019) plus a list of common contaminants and all the corresponding decoy entries (*Choi et al., 2011*). For peptide identification a precursor ion mass tolerance of 7 ppm was used for MS1 level, trypsin was chosen as enzyme, and up to three missed cleavages were allowed. The fragment ion mass tolerance was set to 0.5 Da for MS2 spectra. Oxidation of methionine and N-terminal protein acetylation were used as variable modifications whereas carbamidomethylation on cysteines was set as a fixed modification. False discovery rate in peptide identification was set to a maximum of 5%. The analysis of specific chromatin interactors was carried out with SAINT (v2, Significance Analysis of INTeractome) as previously described (*Choi et al., 2011*; *Perkins et al., 1999*). Replicate 2 of the 2C$^+$ condition was excluded for SAINT analysis due to abnormal lower peptide spectrum matches observed in this run. Hierarchical clustering of all the chromatome replicates was computed and visualized using Instant Clue (*Nolte et al., 2018*) v0.5.2 (http://www.instantclue.uni-koeln.de/). PCC were calculated using the Prism software (v9.0, GraphPad, San Diego, CA, USA). To identify proteins shared by the 2C$^-$ and Luc chromatomes and not enriched in the 2C$^+$ chromatome, an average enrichment value was computed from the respective pairwise comparisons (i.e. Luc vs 2C$^-$; Luc vs 2C$^+$; 2C$^-$ vs Luc; 2C$^-$ vs 2C$^+$) and then selecting those hits that were more commonly enriched among the 2C$^-$ and Luc chromatomes (fold change [FC]≥2). GO term enrichment was performed with GO Enrichment Analysis using the PANTHER tool (*Mi et al., 2019*; *Ashburner et al., 2000*) (https://geneontology.org/). Protein interaction data were retrieved from the STRING database v11.0 (*Szklarczyk et al., 2017*) and visualized with Cytoscape v3.8.2 (*Shannon et al., 2003*).

## **WB analysis**

Protein extracts were boiled in Laemmli buffer, run in precast protein gel (Mini-PROTEAN TGX; 4561084, Bio-Rad), and then transferred to immuoblot polyvinylidene difluoride membranes (162-0177, Bio-Rad). The membranes were blocked and incubated with the indicated primary antibodies overnight at 4°C (rabbit anti-histone H3 [1:1000; ab1791, Abcam], rabbit anti-histone H3K9me3 [1:500; ab8898, Abcam], mouse anti-OCT4 [1:500; sc-5279, Santa Cruz], and mouse anti-SMARCAD1 [1:500; ab67548, Abcam]).

After washing, membranes were incubated with specific peroxidase-conjugated secondary antibodies (sheep anti-mouse IgG HRP-linked [1:1000; NA931, GE Healthcare] and donkey anti-rabbit IgG HRP-linked [1:2000; NA934, GE Healthcare]) and visualized on an Amersham Imager 600 (29083461, GE Healthcare Life Sciences).

**Table 3.** List of top oligos used for cloning single guide RNAs (sgRNAs).

| sgRNA name | Genomic sequence | Strand | sgRNA target sequence | PAM |
|---|---|---|---|---|
| sgSmarcad1 #1 | NC_000072.6 | Antisense | AACAGAGCACATTTAAACTG | GGG |
| sgSmarcad1 #2 | NC_000072.6 | Sense | AGTCTGTAAAACAGCCGCGA | GGG |
| sgTopbp1 #1 | NC_000075.6 | Sense | GAAGCAGAGTGAGCTCAATG | GGG |
| sgTopbp1 #2 | NC_000075.6 | Antisense | GTGATTTGCTAAGAATACCA | AGG |
| sgLuciferase | | | ACAACTTTACCGACCGCGCC | |

## Dot blot analysis

Samples were spotted in triplicates in 1 µl dots onto a nitrocellulose membrane (0.2 µM, Amersham Protan), air-dried, and detected following standard blotting procedures with the corresponding antibodies (rabbit anti-histone H3 [1:1000; ab1791, Abcam], mouse anti-vinculin [1:1000; V9131, Merck]). Quantification of dot blots was performed by Image Studio Lite software (v5.2, LI-COR, Biosciences). For quantification, each protein was normalized to its background signal.

## CRISPR-Cas9 plasmid generation and delivery

sgRNAs targeting each of the specific target genes were retrieved from the Mouse CRISPR Knockout Pooled Library (Addgene #73632). Two sgRNA sequences were selected per gene of interest (for sgRNAs sequences, see *Table 3*). The sgRNAs with the highest on-target activity score (Rule Set 2) were selected for assembly into the CRISPR-Cas9 vector. An sgRNA targeting the luciferase sequence was also included as control. Primers containing sequences for the sgRNAs were annealed in the presence of T4 ligation buffer (Thermo Fisher) and T4 PNK (NEB) in a heat block (30°C for 30 min, 95°C for 5 min and slow cool down to RT). Annealed primers were then cloned into the pU6-(BbsI)_CBh-Cas9-T2A-mCherry plasmid following a one-step cloning reaction. pU6-(BbsI)_CBh-Cas9-T2A-mCherry was a gift from Ralf Kuehn (Addgene plasmid #64324).

To generate CRISPR-Cas9-targeted ESCs, cells were nucleofected with 4 µg of the sgRNA-containing plasmid individually following the Amaxa Mouse ES cell Nucleofector kit recommendations (VPH-1001, Lonza). Later, ESCs were FACS-sorted 48 hr after nucleofection to enrich for the modified cells.

## Zygote collection and culture

Embryos were collected at E0.5 from 6 to 10 weeks' BDF1 female mice (Charles River Laboratories) following 5 IU pregnant mare's serum gonadotrophin and 5 IU human chorionic gonadotropin (hCG) injections at 48 hr intervals. Female mice were mated with BDF1 male mice immediately after hCG injection. Embryos were collected from the oviducts 24 hr post-hCG and were briefly cultured in M2 medium supplemented with 0.2 mg/ml hyaluronidase (H3506, Sigma) to remove cumulus cells. Cumulus-free embryos were washed and cultured with Advanced KSOM medium (MR-101-D, Millipore) at 37°C until microinjection.

## Microinjection of MOs

MOs for *Smarcad1* and *Topbp1* and non-targeting control were designed and produced by Gene Tools (Gene Tools, LLC). MOs were microinjected into the cytoplasm of E0.5 embryos using a Narishige micromanipulator system mounted on an Olympus IX71 inverted microscope. Embryos were immobilized using a holding pipette and MOs were then microinjected using a Narishige pneumatic microinjector (IM-300, Narishige). After microinjection, embryos were cultured in Advanced KSOM medium in low oxygen conditions (5% $CO_2$, 5% $O_2$) at 37°C for 5 days (until E5.5). Preimplantation development was examined every 24 hr using an AMG EVOS microscope.

The following MO sequences were used:

> Control MO: TCCAGGTCCCCCGCATCCCGGATCC;
> *Smarcad1* MO: ATATTGGGAGGAACCACCACCCTGA;
> *Topbp1* MO: ACGGCTCTTGGTCATTTCTGGACAT;

All morpholino sequences are written from 5′ to 3′ and they are complementary to the translation-blocking target.

All animal experiments were approved and performed in accordance with the institutional guidelines (Parc de Recerca Biomèdica de Barcelona [PRBB], Barcelona, Spain) and in accordance with the Ethical Committee for Animal Experimentation (CEEA) number PC-17-0019-PI, approved by La Comissió d'Experimentació Animal, Departament de Territori i Sostenibilitat, Direcció General de Polítiques Ambientals i Medi Natural, Generalitat de Catalunya.

## Statistical analysis

As specified in the figure legends, data are presented either as scatter dot plots with line at mean ± SD or at median ± interquartile range, bar graphs showing mean ± SD, min to max boxplots with

line at median, or as violin plots showing median and quartiles. All statistical tests and graphs were generated using the Prism software (v9.0, GraphPad, San Diego, CA, USA), unless otherwise indicated. Depending on the experimental setup, we used unpaired two-tailed Student's $t$-test, multiple $t$-test, Fisher's exact test, Mann-Whitney test, one-way ANOVA or two-way ANOVA with the indicated post-comparison test. In all cases, a p value $p \leq 0.05$ was considered significant ($p \leq 0.05$*; $p \leq 0.01$**; $p \leq 0.001$***; $p \leq 0.0001$****; $p > 0.05$[ns], not significant). All experiments were replicated in at least three independent biological replicates unless otherwise indicated. The number of independent biological replicates for each experiment is indicated in the figure legends.

No statistical method was used to predetermine sample size in in vitro and in vivo experiments, but group sizes were determined based on the results of preliminary experiments. Group allocation was performed in a randomized fashion. The investigators were not blinded to allocation during outcome assessment.

## Acknowledgements

We would like to thank M-E Torres-Padilla (Helmholtz Zentrum München), BR Cairns (Huntsman Cancer Institute), and B Huang (UCSF) for sharing E14 ESCs containing the 2C::EGFP reporter, an independent E14 ESC clone containing the Dux-CA cassette and Insight3 software. We are grateful to S Sdelci for critical reading of the manuscript. We also thank the CRG/UPF Flow Cytometry Unit, the CRG Advanced Light Microscopy Unit, and the PRBB animal facility (PRBB, Barcelona). This work was supported by the European Union's Horizon 2020 Research and Innovation Programme (No 686637 and No 964342 to MPC), Ministerio de Ciencia e Innovación (grant no. PID2020-114080GB I00/AEI/10.13039/501100011033 and grant no. BFU2017-86760-P/AEI/FEDER, UE to MPC), an AGAUR grant from the Departament de Recerca i Universitats de la Generalitat de Catalunya (2021-SGR2021-01300 to MPC), Fundació La Marató de TV3 (202027-10 to MPC), and National Natural Science Foundation of China (No 31971177 to MPC); Spanish Ministry of Economy, Industry and Competitiveness (MEIC) (PID2019-108322GB-100 to LDC), and from AGAUR (LDC). We acknowledge the support of the Spanish Ministry of Science and Innovation to the EMBL partnership, the Centro de Excelencia Severo Ochoa and the CERCA Programme. The CRG/UPF Proteomics Unit is part of the Spanish Infrastructure for Omics Technologies (ICTS OmicsTech) and it is a member of the ProteoRed PRB3 consortium which is supported by grant PT17/0019 of the PE I+D+i 2013–2016 from the Instituto de Salud Carlos III (ISCIII) and ERDF. RS-P was supported by a FI-AGAUR PhD fellowship from the Secretaria d'Universitats i Recerca del Departament d'Empresa i Coneixement de la Generalitat de Catalunya and the co-finance of Fondo Social Europeo (2018FI_B_00637 and FSE). XT is supported by an FPI PhD fellowship from the Ministerio de Ciencia e Innovación (PRE2018-085107). SA is funded by the Ramon y Cajal program of the Ministerio de Ciencia, Innovación y Universidades, and the European Social Fund under the reference number RYC-2018-025002-I, and the Instituto de Salud Carlos III-FEDER (PI19/01814). LM is supported by a grant for the recruitment of early-stage research staff FI-2020 (Operational Program of Catalonia 2014–2020 CCI grant no. 2014ES05SFOP007 of the European Social Fund) and La Caixa Foundation fellowship (LCF/BQ/DR20/11790016). MP was supported by a Severo Ochoa PhD fellowship from the Subprograma Estatal de Formación del Ministerio de Economía y Competitividad (BES-2015-072802). MVN is funded by FP7/2007–2013 under an REA grant (608959) and Juan de la Cierva-Incorporación 2017.

## Additional information

### Funding

| Funder | Grant reference number | Author |
| --- | --- | --- |
| Horizon 2020 Framework Programme | 686637 | Maria Pia Cosma |
| Horizon 2020 Framework Programme | 964342 | Maria Pia Cosma |

| Funder | Grant reference number | Author |
| --- | --- | --- |
| Ministerio de Ciencia e Innovación | PID2020-114080GB I00/ AEI/10.13039/501100011033 | Maria Pia Cosma |
| Ministerio de Ciencia e Innovación | BFU2017-86760-P/AEI/ FEDER | Maria Pia Cosma |
| Agència de Gestió d'Ajuts Universitaris i de Recerca | 2021-SGR2021-01300 | Maria Pia Cosma |
| Fundació la Marató de TV3 | 202027-10 | Maria Pia Cosma |
| National Natural Science Foundation of China | 31971177 | Maria Pia Cosma |
| Ministerio de Economía y Competitividad | PID2019-108322GB-100 | Luciano Di Croce |
| Instituto de Salud Carlos III | PT17/0019 | Eduard Sabidó |
| Departament d'Empresa i Coneixement, Generalitat de Catalunya | 2018FI_B_00637 | Ruben Sebastian-Perez |
| Ministerio de Ciencia e Innovación | PRE2018-085107 | Xiaochuan Tu |
| Ministerio de Ciencia, Innovación y Universidades | RYC-2018-025002-I | Sergi Aranda |
| Instituto de Salud Carlos III | PI19/01814 | Sergi Aranda |
| Ministerio de Ciencia e Innovación | 2014ES05SFOP007 | Laura Martin |
| 'la Caixa' Foundation | LCF/BQ/DR20/11790016 | Laura Martin |
| Ministerio de Economía y Competitividad | BES-2015-072802 | Martina Pesaresi |
| FP7 People: Marie-Curie Actions | 608959 | Maria Victoria Neguembor |
| Ministerio de Ciencia e Innovación | Juan de la Cierva-Incorporación 2017 | Maria Victoria Neguembor |

The funders had no role in study design, data collection and interpretation, or the decision to submit the work for publication.

## Author contributions
Ruben Sebastian-Perez, Conceptualization, Formal analysis, Investigation, Methodology, Writing - original draft, Writing - review and editing; Shoma Nakagawa, Xiaochuan Tu, Formal analysis, Investigation, Methodology; Sergi Aranda, Martina Pesaresi, Pablo Aurelio Gomez-Garcia, Marc Alcoverro-Bertran, Jose Luis Gomez-Vazquez, Davide Carnevali, Eva Borràs, Eduard Sabidó, Laura Martin, Malka Nissim-Rafinia, Eran Meshorer, Investigation; Maria Victoria Neguembor, Luciano Di Croce, Supervision, Writing - original draft; Maria Pia Cosma, Conceptualization, Formal analysis, Supervision, Funding acquisition, Writing - original draft, Project administration, Writing - review and editing

## Author ORCIDs
Ruben Sebastian-Perez https://orcid.org/0000-0001-7209-7612
Sergi Aranda https://orcid.org/0000-0003-3853-430X
Maria Pia Cosma https://orcid.org/0000-0003-4207-5097

## Ethics
All animal experiments were approved and performed in accordance with institutional guidelines [Parc de Recerca Biomèdica de Barcelona (PRBB), Barcelona, Spain] and in accordance with the Ethical Committee for Animal Experimentation (CEEA) number PC-17-0019-PI, approved by La Comissió d'Experimentació; Animal, Departament de Territori i Sostenibilitat, Direcció General de Polítiques Ambientals i Medi Natural, Generalitat de Catalunya.

Reviewer #1 (Public review): https://doi.org/10.7554/eLife.87742.3.sa1
Reviewer #2 (Public review): https://doi.org/10.7554/eLife.87742.3.sa2
Reviewer #3 (Public review): https://doi.org/10.7554/eLife.87742.3.sa3
Author response https://doi.org/10.7554/eLife.87742.3.sa4

## Additional files

### Supplementary files
Supplementary file 1. Spreadsheet containing proteomic data. (A) List of protein groups identified by mass spectrometry in –EdU, Luc, 2C⁻ and 2C⁺ cells. (B) Input data for Significance Analysis of INTeractome (SAINT) analysis. (C) SAINT results of the comparison 2C⁻ vs Luc and 2C⁺ vs Luc. (D) SAINT results of the comparison Luc vs 2C⁻ and 2C⁺ vs 2C⁻. (E) SAINT results of the comparison Luc vs 2C⁺ and 2C⁻ vs 2C⁺. (F) Total number of peptide spectrum matches (PSM) per protein in the different cells and conditions tested.

MDAR checklist

### Data availability
The mass spectrometry proteomics data have been deposited to the ProteomeXchange Consortium via the PRIDE partner repository with the dataset identifier PXD019703. All other data needed to evaluate the conclusions in this study are present in the paper and/or the supplementary files. Additional materials generated in this study are available from the corresponding author upon request.

The following dataset was generated:

| Author(s) | Year | Dataset title | Dataset URL | Database and Identifier |
| --- | --- | --- | --- | --- |
| Sebastian-Perez R | 2025 | SMARCAD1 and TOPBP1 contribute to heterochromatin maintenance at the transition from the 2C-like to the pluripotent state | https://proteomecentral.proteomexchange.org/cgi/GetDataset?ID=PXD019703 | ProteomeXchange, PXD019703 |

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
