## [Editor Report · eLife Assessment]

This **important** study examines heterochromatin domain dynamics using a model system that allows reversible transition from an embryonic stem cell to a 2-cell-like state. The authors present a **solid** resource to the research community that will further the understanding of changes in the chromatin-bound proteome during the 2C-to-ESC transition. However, conclusions related to the functional roles of the interaction between the SWI/SNF complex component SMARCAD1 and the DNA Topoisomerase II Binding protein (TOPBP1) remain incomplete.

---

## [Referee Report · Reviewer #1 (Public review)]

In this study, the authors investigate the molecular mechanisms driving the establishment of constitutive heterochromatin during embryonic development. The experiments have been meticulously conducted and effectively address the proposed hypotheses.

The methodology stands out for its robustness, utilizing:

i) an efficient system for converting ESCs to 2C-like cells via Dux overexpression;

ii) a global approach through IPOTD, which unveils the chromatome at distinct developmental stages; and

iii) STORM technology, enabling high-resolution visualization of DNA decompaction. These tools collectively provide clear and comprehensive insights that support the study's conclusions.

The work makes a significant contribution to the field, offering valuable insights into chromatin-bound proteins at critical stages of embryonic development. These findings may also inform our understanding of processes beyond heterochromatin maintenance.

The revised manuscript shows improvement, particularly through enhanced discussion and the addition of new references addressing the cooperation of SMARCAD1 and TOPBP1. All my previous concerns have been thoroughly addressed by the authors. However, I believe that, as this reviewer suggested, the inclusion of a model that summarizes the main findings of the study and discusses the potential mechanisms involved, would enhance the clarity and understanding of the message the manuscript aims to convey.

---

## [Referee Report · Reviewer #2 (Public review)]

As noted in the original review, the study by Sebastian-Perez addresses an important research question using a tractable model system to examine the earliest drivers of heterochromatin formation during embryogenesis. Moreover, the proteomic analyses provide a valuable resource to the research community to understand changes in the chromatin-bound proteome during the 2C-to-ESC transition. From there, they carry out more detailed analyses of TOPBP1, which shows substantive changes in chromatin association in 2C-like cells, and a potential interacting protein SMARCAD1, which shows only modest changes in chromatin association. While I appreciate that the authors have revised the manuscript to some extent to address the minor points raised, the major over-arching issue of how TOPBP1 and SMARCAD1 function in the 2C-like state is still a concern.

---

## [Referee Report · Reviewer #3 (Public review)]

The manuscript entitled "SMARCAD1 and TOPBP1 contribute to heterochromatin maintenance at the transition from the 2C-like to the pluripotent state" by Sebastian-Perez et al. adopted the iPOTD method to compare the chromatin-bound proteome in ESCs and 2CLCs induced by Dux overexpression. The authors identified 397 chromatin-bound proteins enriched specifically in non-2CLCs, among which they further investigated TOPBP1 due to its potential role in chromocenter reorganization. SMARCD1, a known interacting protein of TOPBP1, was also investigated in parallel. The authors report increased size and decreased number of H3K9me3-heterochromatin foci in Dux-induced 2CLCs. Remarkably, depletion of either TOPBP1 or SMARCD1 resulted in similar phenotypes. However, the absence of these proteins did not affect the entry into or exit from the 2C-like state. The authors further showed that both TOPBP1 and SMARCD1 are essential for early embryonic development.

This manuscript provides valuable insights into the features of 2CLCs regarding H3K9me3-heterochromatin reorganization. However, the findings are largely descriptive. Mechanistic studies are required in future studies, such as: (1) how SMARCD1 associates with H3K9me3 and contributes to heterochromatin maintenance, (2) how TOPBP1 regulates the expression of SMARCD1 and facilitates its localization in heterochromatin foci, (3) whether the remodelling of chromocenter directly influence the transitions between ESCs and 2CLCs.

---

## [Author Response]

The following is the authors’ response to the original reviews.

**Reviewer #1 (Public Review):**
In the present work the authors explore the molecular driving events involved in the establishment of constitutive heterochromatin during embryo development. The experiments have been carried out in a very accurate manner and clearly fulfill the proposed hypotheses.Regarding the methodology, the use of: (i) an efficient system for conversion of ESCs to 2C-like cells by Dux overexpression; (ii) a global approach through IPOTD that reveals the chromatome at each stage of development and (iii) the STORM technology that allows visualization of DNA decompaction at high resolution, helps to provide clear and comprehensive answers to the conclusion raised.The contribution of the present work to the field is very important as it provides valuable information on chromatin-bound proteins at key stages of embryonic development that may help to understand other relevant processes beyond heterochromatin maintenance.The study could be improved through a more mechanistic approach that focuses on how SMARCAD1 and TOPBP1 cooperate and how they functionally connect with H3K9me3, HP1b and heterochromatin regulation during embryonic development. For example, addressing why topoisomerase activity is required or whether it connects (or not) to SWI/SNF function and the latter to heterochromatin establishment, are questions that would help to understand more deeply how SMARCAD1 and TOPBP1 operate in embryonic development.

We would like to thank the reviewer for the positive evaluation of our work and the methodology we employed. We greatly appreciated the reviewer’s recognition of our study to “provide valuable information on chromatin-bound proteins at key stages of embryonic development that may help to understand other relevant processes beyond heterochromatin maintenance”. While we acknowledge the value of including mechanistic studies, such an addition would require a substantial amount of experimental work that exceeds our current resources.

**Reviewer #1 (Recommendations For The Authors):**
In my opinion, the authors could improve the study by deciphering -to a certain extent- the possible mechanism by which SMARCAD1 and TOPBP1 are cooperating in their system to establish H3K9me3 and consequently heterochromatin; and whether it is different (or not) from that already reported in yeast (ref 27). In fact, is it only SMARCAD1 that participates in this process or the whole SWI/SNF complex? Could the lack of SMARCAD1 compromise the proper assembly of the SWI/SNF complex? In this regard, a model describing the main findings of the study and the discussion of the possible mechanisms involved -based on the current bibliography- would be appreciated. This, although speculative, would illustrate the range of possibilities that could be operating in the maintenance of heterochromatin during embryonic development. In conclusion, it would be great if the authors could link -mechanistically- the dots connecting SMARCD1, TOPBP1, H3K9me3/HP1/heterochromatin.

As suggested by the reviewer and to enrich the discussion, we have included some additional sentences and references in the revised discussion section.

As a minor point, In Figure 3A, left panel it appears that the protein precipitating with H3K9me3 reacts with TOPBP1 but its molecular weight does not exactly match to the TOPBP1 band found in the input. The authors should clarify this point and it is also recommended that IPs and inputs are run in the same gel. Please replace Figure 3A right panel.

Following the reviewer’s suggestion and to improve the reading flow, we have restructured the order of the figures and removed the original Figure 3A. The revised Figure 3A-C panel illustrates the SMARCAD1 association with H3K9me3 in ESCs and 2C- cells, while capturing the reduced SMARCAD1-H3K9me3 association in 2C^+^ cells.

**Reviewer #2 (Public Review):**
The manuscript by Sebastian-Perez describes determinants of heterochromatin domain formation (chromocenters) at the 2-cell stage of mouse embryonic development. They implement an inducible system for transition from ESC to 2C-like cells (referred to as 2C^+^) together with proteomic approaches to identify temporal changes in associated proteins. The conversion of ESCs to 2C^+^ is accompanied by dissolution of chromocenter domains marked by HP1b and H3K9me3, which reform upon transition back to the 2C-like state. The innovation in this study is the incorporation of proteomic analysis to identify chromatin-associated proteins, which revealed SMARCAD1 and TOPBP1 as key regulators of chromocenter formation.In the model system used, doxycycline induction of DUX leads to activation of EGFP reporter regulated by the MERVL-LTR in 2C^+^ cells that can be sorted for further analysis. A doxycycline-inducible luciferase cell line is used as a control and does not activate the MERVL-LTR GFP reporter. The authors do see groups of proteins anticipated for each developmental stage that suggest the overall strategy is effective.The major strengths of the paper involve the proteomic screen and initial validation. From there, however, the focus on TOPBP1 and SMARCAD1 is not well justified. In addition, how data is presented in the results section does not follow a logical flow. Overall, my suggestion is that these structural issues need to be resolved before engaging in comprehensive review of the submission. This may be best achieved by separating the proteomic/morphological analyses from the characterization of TOPBP1 and SMARCAD1.

We appreciate the reviewer’s positive evaluation of our inducible system to trigger the transition from ESCs to 2C-like cells, and the strength of the chromatin proteomics we conducted. In response to the reviewer’s suggestion, we have reorganized the order of the figures, particularly Figure 1 and Figure 2, and revised the text to improve readability and flow.

**Reviewer #2 (Recommendations For The Authors):**
There are some very interesting components to the study but, as noted, the narrative requires changes and the rationale for focusing on TOPBP1 and SMARCAD1 is not strong at present. Specific comments are noted below(1) Inclusion of authentic 2C cells for comparative chromocenter analysis (or at least a more fulsome discussion of how the system has been benchmarked in previous studies).

We have included more detail in the revised methods section, in the “Cell lines and culture conditions” paragraph. We have added: “The Dux overexpression system was benchmarked according to previously reported features. Dux overexpression resulted in the loss of DAPI-dense chromocenters and the loss of the pluripotency transcription factor OCT4 (fig. S1E) (6, 7), upregulation of specific genes of the 2-cell transcriptional program such as endogenous Dux, MERVL, and major satellites (MajSat) (fig. S1F) (6, 7, 11, 26, 58), and accumulation in the G2/M cell cycle phase (fig. S1G), with a reduced S phase consistent in several clonal lines (fig. S1H) (15).”

(2) In Figure 1A, the text indicates a loss of chromocenters, but it may be better described as decompaction because the DAPI/H3K9me3 staining shows diffuse/expanded structures (this is in fact how it is described in relation to Figure 2).

We have changed the text accordingly, now describing it as “decompaction”.

(3) Table S1 has 6 separate tabs but these are not specified in the text. It would be useful to separate the 397 proteins unique to Luc and 2C- cells since they form much of the basis for the remaining analysis. This approach also assumes it is the absence of a protein in the 2C^+^ that accounts for the lack of chromocenters (noting there are 510 proteins unique to the 2C^+^ state that are not discussed).

We have referenced the supplementary table as Table S1 in the text for simplicity. It includes: Table S1A - List of Protein Groups identified by mass spectrometry in -EdU, Luc, 2C- and 2C^+^ cells; Table S1B - Input data for SAINT analysis; Table S1C - SAINT results of the comparison 2C- vs Luc and 2C^+^ vs Luc; Table S1D - SAINT results of the comparison Luc vs 2C- and 2C^+^ vs 2C-; Table S1E - SAINT results of the comparison Luc vs 2C^+^ and 2C- vs 2C^+^; and Table S1F - Total number of PSM per protein in the different cells and conditions tested.

(4) Since there is no change in H3K9me3 levels, loss of SUV420H2 from 2C^+^ chromatin (figure 1G) coupled with potential changes in H4K20me3 could contribute the morphological differences. SUV420H2 is known to regulate chromocenter clustering in a way the requires H4K20me3 but this is not addressed or cited (PUBMED: 23599346).

As suggested by the reviewer, we have added additional sentences and references in the revised manuscript.

(5) In Figure 1C, there does appear to be overlap between the 2C^+^ and 2C- populations (while the Luc population is distinct) even though they are morphologically distinct when imaged in Figure 2A. The 2C- cells are thought to be an intermediate, low Dux expressing population.

Chromatome profiling through genome capture provides a snapshot of the chromatin-bound proteome in the analyzed samples (shown in revised Fig. 2B). As indicated by the reviewer and previously reported in the literature, 2C- cells are an intermediate population before reaching 2C^+^ cells. For this study, we have focused on H3K9me3 morphological changes. Even though 2C- and 2C^+^ cells are distinct with respect to H3K9me3 morphology (shown in revised Fig. 1B), analysis of the chromatome data from hundreds of chromatin-bound proteins revealed some overlap between these two populations. However, replicates from the same population tend to cluster together, for example, 2C^+^ rep1 and 2C^+^ rep3, and 2C- rep1 and 2C- rep2. Collectively, these data suggest that a defined subset of coordinated changes in the chromatome likely triggers the transition from 2C- to 2C^+^ cells. Further experimental investigation of the chromatome dataset during the 2C-like transition would be interesting, however, we believe it is beyond the scope of this study.

(6) Data with SUV39H1 and 2 is difficult to accommodate; what about other H3K9 methyltransferases or proteins such as TRIM28 (KAP1) and SETDB1 (this comes up in the discussion but is not assessed in the results section).

We agree that investigating the role of TRIM28 (KAP1) and SETDB1 in this experimental setting could be of interest, however, we believe that these experiments go beyond the scope of the presented study.

(7) Rationale for choosing TOPBP1 needs to be improved. How do TOPBP1 levels relate to TOPI/TOP2A/TOP2B levels across the 3 cell populations? By what criteria does topoisomerase inhibitor treatment increase 2C^+^ like cells? Moreover, to what extent will inhibiting topoisomerases lead to global heterochromatin and cell cycle changes regardless of cell type.

Following the reviewer’s suggestion, we have included some additional references throughout the text to strengthen our rationale for selecting TOPBP1, given its well-established critical role in DNA replication and repair. Additionally, we have revised the results and discussion sections to include new sentences that propose a potential mechanism by which topoisomerase inhibitors may indirectly recruit TOPBP1 to facilitate DNA repair, ultimately leading to an increase in 2C^+^ cells.

(8) Likewise, the decision to look at SMARCAD1 based solely on its interaction with TOPBP1 seems somewhat arbitrary and it did not seem to come up as of interest in the iPOTD analysis. Moreover, they were not able to validate the interaction with their own analyses.

We have revised the text to clarify the connection further.

(9) The flow of results is confusing. The first section concludes with a focus on TOPBP1 and SMARCAD1, then progresses to morphological characterization of heterochromatin regions in the next two sections before returning to TOPBP1 and SMARCAD1. It seems like it would make more sense to describe the model system and morphological characterization at the beginning of the results section and then transition to the proteomic analysis and characterization of TOPBP1 and SMARCAD1 (with the expectation that the rationale be improved).

As suggested by the reviewer, we have reordered the figures, particularly Figure 1 and Figure 2, and rephased the text to improve the overall reading flow.

(10) There has been considerable work done on characterizing chromatin structure, epigenetic changes, and morphology during early embryonic development. It is therefore difficult to see what validating some of these changes in the inducible model is adding much in the way of new knowledge. It may, but this is not articulated in the current text.

As detailed before, we have rephrased the text to improve the overall reading flow, which we hope has improved the understanding of the impact of our results.

(11) It is difficult to disentangle broader effects of both TOPBP1 and SMARCAD1 from those described here; they may induce phenotypes, but these may not be unique to this model system.

We agree with the reviewer, but to address this point would require additional experiments which would go beyond the scope of the presented study.

(12) One of the issues with this assay is global chromatin recovery; it is not focused on heterochromatin compartments. The statement "We identified a total of 2396 proteins, suggesting an efficient pull-down of chromatin-associated factors (fig. S2D and Table S1)" does not demonstrate efficiency. Additional functional annotation would be required to establish this claim, including what fraction are known chromatin-associated proteins (with a focus on the heterochromatin compartment).

We have changed the text accordingly. The resulting statement reads as: “We identified a total of 2396 proteins, suggesting an effective pull-down of putative chromatin-associated factors (fig. S2D and Table S1)”.

**Reviewer #3 (Public Review):**
The manuscript entitled "SMARCAD1 and TOPBP1 contribute to heterochromatin maintenance at the transition from the 2C-like to the pluripotent state" by Sebastian-Perez et al. adopted the iPOTD method to compare the chromatin-bound proteome in ESCs and 2C-like cells generated by Dux overexpression. The authors identified 397 chromatin-bound proteins enriched only in ESC and 2C- cells, among which they further investigated TOPBP1 due to its potential role in controlling chromocenter reorganization. SMARCD1, a known interacting protein of TOPBP1, was also investigated in parallel. The authors observed increased size and decreased number of H3K9me3-heterochromatin foci in Dux-induced 2C^+^ cells. Interestingly, depletion of TOPBP1 or SMARCD1 also led to increased size and decreased number of H3K9me3 foci. However, depletion of these proteins did not affect entry into or exit from the 2C-like state. Nevertheless, the authors showed that both TOPBP1 and SMARCD1 are required for early embryonic development.Although this manuscript provides new insights into the features of 2C-like cells regarding H3K9me3-heterochromatin reorganization, it remains largely descriptive at this stage. It does not provide new insights into the following important aspects: (1) how SMARCD1 associates with H3K9me3 and contributes to heterochromatin maintenance, (2) how TOPBP1 regulates the expression of SMARCD1 and facilitates its localization in heterochromatin foci, (3) whether the remodelling of chromocenter is causally related to the mutual transitions between ESCs and 2C-like cells. Furthermore, some results are over-interpreted. Additional experiments and analyses are needed to increase the strength of mechanistic insights and to support all claims in the manuscript.

We would like to thank the reviewer for their positive and thorough evaluation of our manuscript. We have revised the text and hope that the overall flow is now clearer. Moreover, while we acknowledge the value of including mechanistic studies, such an addition would require a substantial amount of experimental work that exceeds our current resources.

**Reviewer #3 (Recommendations For The Authors):**
Major points:(1) Fig.2: the DNA decompaction of the chromatin fibers shown in 2C^+^ cells may be more related to a relaxed 3D chromatin conformation (Zhu, NAR 2021; Olbrich, Nat Commun 2021) than chromatin accessibility. The authors should discuss this point.

As suggested by the reviewer, we have included some additional sentences and references in the revised manuscript to address this concern.

(2) Chemical inhibition of topoisomerases resulted in an increase in the percentage of 2C^+^ cells. Does depletion of TOPBP1 also resulted in increased percentage of 2C^+^ cells? Please include this result in Fig. 3E. Additionally, it should be noted that DDR and p53 have been reported to activate Dux (Stashpaz, eLife 2020; Grow, Nat Genet 2021), and thus, may contribute to the increased percentage of 2C^+^ cells observed upon topoisomerase inhibition. This point should be discussed in the manuscript.

To address this concern, we have included some additional sentences and references in the revised manuscript.

(3) Fig 3A: the TOPBP1 band in the IP sample is questionable, and therefore the conclusion that TOPBP1 is associated with H3K9me3 is difficult to draw from Fig 3A. Additionally, the authors mentioned that association of TOPBP1 and SMARCAD1 is undetected in ESCs, likely due to the suboptimal efficiency of available antibodies. As these are key conclusions in this study, the authors are suggested to try other commercially available TOPBP1 antibodies (e.g., Abcam #ab-105109, used by ElInati, PNAS 2017) or knock-in tags to perform the co-IP experiment.

Following the reviewer’s suggestion and to improve the reading flow, we have restructured the order of figures and removed the original Figure 3A. The revised Figure 3A-C panel illustrates the SMARCAD1 association with H3K9me3 in ESCs and 2C- cells, while capturing the reduced SMARCAD1-H3K9me3 association in 2C^+^ cells.

(4) Fig. 3C-D, Fig. S3D: the authors claimed reduction of both SMARCAD1 expression and its co-localization with H3K9me3 foci in 2C^+^ cells, but did not perform mechanistic studies. It is important to know if TOPBP1 expression also decreases in 2C^+^ cells. Additionally, it is unclear if the reduced co-localization of SMARCAD1 with H3K9me3 foci results from its altered nuclear localization or simply from reduced expression level? In either case, please provide some mechanistic insights.

While we acknowledge the value of including mechanistic studies, such an addition would require a substantial amount of experimental work that exceeds our current resources.

(5) Fig. 3K, Fig. S4D-E: does SMARCAD1 expression decrease upon TOPBP1 depletion? Statistical analysis of SMARCAD1 intensity in Fig. S4E is needed, and a Western blot analysis is strongly suggested. Additionally, it is unclear if the reduced co-localization of SMARCAD1 with H3K9me3 foci results from its altered nuclear localization or simply from reduced expression level? In Fig. 3K, TOPBP1-depleted cells appear to show decreased size and increased number of H3K9me3 foci, which is inconsistent with Fig. S4B-C. The authors should clarify this discrepancy. Furthermore, statistics should be performed to determine whether Smarcad1/Topbp1 knockdown could further increase the size and decrease the number of H3K9me3 foci in 2C^+^ cells. This would provide additional evidence for the involvement of these proteins in heterochromatin maintenance.

We did not observe Smarcad1 downregulation after Topbp1 knockdown (shown in fig. S4A). In Figs. S4B and S4C, we observed that the number of H3K9me3 foci decreased, and their area became larger after knocking down either Smarcad1 or Topbp1, compared to scramble controls. These results align with the reviewer’s comment. Additionally, it should be noted that these findings were derived from the quantification of tens of cells and hundreds of foci, as indicated in the figure legend. This resulted in statistical significance after applying the test indicated in the figure legend.

(6) Fig. 3J is suggested to be moved to Fig. 4. Additionally, performing immunostaining of SMARCAD1, TOPBP1, and H3K9me3 during pre-implantation development would provide valuable information on their protein-level dynamics, interactions, and functions in early embryos. This would further strengthen the conclusions drawn in the manuscript.

We agree that performing these additional experiments would provide additional valuable information, however this would require a substantial amount of experimental work that exceeds our current resources.

(7) Fig. 4 and Fig. S5: the authors observed reduced H3K9me3 signal in the Smarcad1 MO embryos at the 8-cell stage, but claim that they failed to examine Topbp1 MO embryos at the 8-cell stage due to their developmental arrest at the 4-cell stage. However, based on Fig. 4A, not all Topbp1 MO embryos were arrested at the 4-cell stage, and it is still possible to examine the H3K9me3 signal in 8-cell Topbp1 MO embryos, which is critical for demonstrating its function in early embryos. Also, how to interpret the increased HP1b signal in Topbp1 MO embryos?

For Topbp1 silencing, we observed an even more severe phenotype compared to Smarcad1 MO. All the Topbp1 MO-injected embryos (100 %) arrested at the 4-cell stage and did not develop further (shown in Fig. 4A and 4B). Therefore, the severity of the Topbp1 morpholino phenotype posed a technical challenge in evaluating the H3K9me3 signal in 8-cell Topbp1 MO embryos, as none of the injected embryos developed beyond the 4-cell stage.

We believe the increased HP1b signal in Topbp1 MO embryos could indicate potential alterations in chromatin organization and heterochromatin stability. Specifically, we observed remodeling of heterochromatin in both 2-cell and 4-cell Topbp1 MO arrested embryos compared to controls, as evidenced by the spreading and increased HP1b signal (shown in fig. S5F-S5I). Further investigations could enhance our understanding of the underlying defects in Topbp1 knockdown embryos, extending beyond heterochromatin-related errors.

Minor points:(1) Page 4, the third row from the bottom: please revise the sentence.

We have reviewed the text and it now reads correctly in the revised manuscript.

(2) Fig. 1C: The authors claimed "Luc replicates clustered separately from 2C^+^ and 2C- conditions", however, Luc rep3 is apparently clustered with 2C conditions.(3) The GFP signal in Fig. S1E is confusing.(4) Please include ESC in Fig. 2D-E. Also label the colors in Fig. 2E.

As indicated in the figure legend of the revised Fig. 1F: “Cells with a GFP intensity score > 0.2 are colored in green. Black dots indicate 2C- cells and green dots indicate 2C^+^ cells.”

(5) Fig. 2G: Transposition of the heatmap (show genes in rows) is suggested to improve readability.(6) Page 7, the third row from the bottom: incorrect citation of Fig. 1K.

Thank you for spotting this incorrect citation. We have corrected it in the revised manuscript.

(7) Page 8, row 15, Fig. S3D should be cited to support the decreased expression of SMARCAD1 in 2C^+^ cells.

We have cited the corresponding supplementary figure S3D in the mentioned sentence.

(8) Fig. 2H: what is the difference between "2C-" and "ESC-like"?

We named 2C- to those cells not expressing the GFP reporter in the transition from ESCs to 2C^+^ cells. We named ESC-like cells to those cells that do not express the GFP reporter during exit, meaning from sorted and purified 2C^+^ to a GFP negative state.

(9) Fig. S4A-C: compared with shTopbp1#2, shTopbp1#1 appears to be slightly more effective in knockdown, but less dramatic changes in the size/number of H3K9me3 foci.(10) Fig. 4: please show the effectiveness of Topbp1 MO by Immunostaining of TOPBP1.(11) Fig. 4C: please label the developmental stage as in Fig. 4E and 4G.

We have added a “8-cell” label in the Figure 4C, as suggested by the reviewer.